# Tight Risk Bounds for
# Gradient Descent on Separable Data

**Matan Schliserman**[1][*]     **Tomer Koren**[1][2]

[1]Blavatnik School of Computer Science, Tel Aviv University
[2]Google Research, Tel Aviv

## Abstract

We study the generalization properties of unregularized gradient methods applied to separable linear classification—a setting that has received considerable attention since the pioneering work of Soudry et al. [14]. We establish tight upper and lower (population) risk bounds for gradient descent in this setting, for any smooth loss function, expressed in terms of its tail decay rate. Our bounds take the form $\Theta(r_{\ell,T}^2/\gamma^2 T + r_{\ell,T}^2/\gamma^2 n)$, where $T$ is the number of gradient steps, $n$ is size of the training set, $\gamma$ is the data margin, and $r_{\ell,T}$ is a complexity term that depends on the tail decay rate of the loss function (and on $T$). Our upper bound greatly improves the existing risk bounds due to Shamir [13] and Schliserman and Koren [12], that either applied to specific loss functions or imposed extraneous technical assumptions, and applies to virtually any convex and smooth loss function. Our risk lower bound is the first in this context and establish the tightness of our general upper bound for any given tail decay rate and in all parameter regimes. The proof technique used to show these results is also markedly simpler compared to previous work, and is straightforward to extend to other gradient methods; we illustrate this by providing analogous results for Stochastic Gradient Descent.

## 1   Introduction

Recently, there has been a marked increase in interest regarding the generalization capabilities of unregularized gradient-based learning methods. One specific area of attention in this context has been the setting of linear classification with separable data, where a pioneering work by Soudry et al. [14] showed that, when using plain gradient descent to minimize the empirical risk on a linearly separable training set with an exponentially-tailed classification loss (such as the logistic loss), the trained predictor will asymptotically converge in direction to the max-margin solution. As a result, standard margin-based generalization bounds for linear predictors suggest that, provided the number of gradient steps ($T$) is sufficiently large, the produced solution will not overfit, despite the lack of explicit regularization and the fact that its magnitude (i.e., Euclidean norm) increases indefinitely with $T$. This result has since been extended to incorporate other optimization algorithms and loss functions [3, 4, 8, 9, 5].

Despite the high interest in this problem, the tight finite-time (population) risk performance of unregularized gradient methods, and even just of gradient descent, have not yet been fully understood. The convergence to a high margin solution exhibited by Soudry et al. [14] (for the logistic loss) occurs at a slow logarithmic rate, thus, the risk bounds for the trained predictors only become effective when $T$ is at least exponentially large in comparison to the size of the training set $n$ and the margin $\gamma$. In a more recent work, Shamir [13] established risk bounds of the form $\widetilde{O}(1/\gamma^2 T + 1/\gamma^2 n)$ for several gradient methods in this setting (still with the logistic loss), that apply for smaller and more realistic values of $T$.

---

[*]Corresponding author; schliserman@mail.tau.ac.il

37th Conference on Neural Information Processing Systems (NeurIPS 2023).

Table 1: Comparison between the bounds established in this work and bounds established in previous work for for gradient descent on $\gamma$-separable data; here, $T$ is the number of gradient steps and $n$ is the size of the training set. Logarithmic factors other than $\log T$ factors are suppressed from the bounds. Examples of how our general bounds are instantiated for some concrete loss functions are presented in Table 2 below.

| Paper | Class of losses | Type of bound | Upper bound | Lower bound |
|---|---|---|---|---|
| Shamir [13] | Logistic loss | 0-1 loss (population) | $\frac{\log^2 T}{\gamma^2 T} + \frac{1}{\gamma^2 n}$ | $\frac{1}{\gamma^2 T}$ (a) |
| Schliserman and Koren [12] | General convex; smooth; Lipschitz; $(c,\delta)$-self-bounded;[(b)] Decay rate of $\phi$ | Risk (population) | $\frac{r_{\phi,T}^2}{\gamma^2 T} + \frac{T^\delta r_{\phi,T}^{2-2\delta}}{\gamma^2 n}$ | – |
| | General convex; smooth; $(c,\delta)$-self-bounded;[(b)] Decay rate of $\phi$ | Risk (population) | $\frac{r_{\phi,T}^2}{\gamma^2 T} + \frac{T^{2\delta} r_{\phi,T}^{4-4\delta}}{\gamma^4 n^{1+2\delta}}$ | – |
| Telgarsky [16] | Convex linear models; quadratically-bounded;[(c)] Decay rate of $\phi$ | Risk (population) | $\frac{r_{\phi,T}}{\gamma \sqrt{T}}$ (d) | – |
| **This paper** | Convex linear models; smooth; Decay rate of $\phi$ | Risk (population) | $\frac{r_{\phi,T}^2}{\gamma^2 T} + \frac{r_{\phi,T}^2}{\gamma^2 n}$ | $\frac{r_{\phi,T}^2}{\gamma^2 T} + \frac{r_{\phi,T}^2}{\gamma^2 n}$ |

[a] The lower bound is shown for the empirical (logistic) risk.
[b] $f$ is $(c,\delta)$-self-bounded if for every $x$, it holds that $\|\nabla f(x)\| \le c f(x)^{1-\delta}$; for $\delta < 1/2$, this property is stronger than smoothness of $f$.
[c] $\ell$ is $(c_1, c_2)$-quadratically-bounded if, for every $x$ and $z$, $|\ell'(x,z)| \le c_1 + c_2(\|x\| + \|z\|)$.
[d] The upper bound is valid when $T \le n$.

Later Schliserman and Koren [12] gave a more general analysis that extends the bounds of Shamir [13] to a wide range of smooth loss functions satisfying a certain "self-boundedness" condition, and Telgarsky [16] provided a test-loss analysis of stochastic mirror descent on "quadratically-bounded" losses. However, all of these works either assume a specific loss (e.g., the logistic loss), impose various conditions on the loss function (beyond smoothness), or do not establish the tightness of their bounds in all regimes of parameters ($T$, $n$ and $\gamma$), and especially in the regime where the number of gradient steps $T$ is larger than the sample size $n$ (see Table 1 for details).

## 1.1 Our contributions

In this work, we close these gaps by showing nearly matching upper and lower risk bounds for gradient descent, which hold essentially for *any* smooth loss function, and for any number of steps $T$ and sample size $n$. Compared to recent prior work in this context [13, 16, 12], our results do not require any additional assumptions on the loss function besides its smoothness, and they strictly improve upon the existing bounds by their dependence on $T$. Further, to the best of our knowledge, risk lower bounds were not previously explored in this context, and the lower bounds we give establish, for the first time, the precise risk convergence rate of gradient descent for any smooth loss function and in all regimes of $T$ and $n$.

In some more detail, our results assume the following form. Let $C_{\phi,\beta}$ be the class of nonnegative, convex and $\beta$-smooth loss functions $\ell(u)$ that decay to zero (as $u \to \infty$) faster than a reference "tail function" $\phi : [0, \infty) \to \mathbb{R}$. The function $\phi$ is merely used to quantify how rapidly the tails of loss functions in $C_{\phi,\beta}$ decay to zero. Then, the risk upper and lower bounds we prove for gradient descent are of the following form:[2]

$$\frac{\beta r_{\phi,T}^2}{\gamma^2 T} + \frac{\beta r_{\phi,T}^2}{\gamma^2 n}. \tag{1}$$

---

[2]For simplicity, we specialize the bounds here to gradient descent with stepsize $\eta = \Theta(1/\beta)$, but the bounds hold more generally to any stepsize smaller than $O(1/\beta)$.

Table 2: Examples of tight population risk bounds established in this paper for gradient descent on $\gamma$-separable data, instantiated for several different loss tail decay rates. Details on the derivation of the bounds from the general Theorems 1 and 2 can be found in Appendix A.

| Tail decay ($\phi$) | $r_{\phi,T}$ | Risk bound |
|---|---|---|
| $\exp(-x)$ | $\log(T)$ | $\Theta\left(\dfrac{\log^2(T)}{\gamma^2 T} + \dfrac{\log^2(T)}{\gamma^2 n}\right)$ |
| $x^{-\alpha}$ | $(\gamma^2 T)^{\frac{1}{\alpha+2}}$ | $\Theta\left(\left(\dfrac{1}{\gamma}\right)^{\frac{2\alpha}{2+\alpha}}\left(\dfrac{1}{T^{\frac{\alpha}{2+\alpha}}} + \dfrac{T^{\frac{2}{2+\alpha}}}{n}\right)\right)$ |
| $\exp(-x^\alpha)$ | $\log^{\frac{1}{\alpha}}(T)$ | $\Theta\left(\dfrac{\log^{\frac{2}{\alpha}}(T)}{\gamma^2 T} + \dfrac{\log^{\frac{2}{\alpha}}(T)}{\gamma^2 n}\right)$ |

The bounds depend on the tail function $\phi$ through the term $r_{\phi,T}$ that, roughly, equals $\phi^{-1}(\varepsilon)$ for $\varepsilon$ chosen such that $\varepsilon \approx (\phi^{-1}(\varepsilon))^2 / \gamma^2 T$ (for the precise statements refer to Theorems 1 and 2, and for concrete examples of implied bounds, see Table 2).

The form of Eq. (1) resembles the bounds given in recent work by Schliserman and Koren [12]. However, the latter work imposed an extraneous "self-boundedness" assumption (stronger than smoothness, see Table 1) that we do not require, and they did not establish the tightness of their bounds, as we do in this paper by providing matching lower bounds. On the flip side, their upper bounds apply in a broader stochastic convex optimization setup, whereas our bounds are specialized to generalized linear models for classification.

In terms of techniques, our proof methodology is also distinctly (and perhaps somewhat surprisingly) simple compared to previous work. Our upper bounds rely on two elementary properties that gradient methods admit when applied to a smooth and realizable objective: low training loss and low norm of the produced solutions. Both properties are obtained from fairly standard arguments and convergence bounds for smooth gradient descent, when paired with conditions implied by the decay rate of the loss function. Finally, to bound the gap between the population risk and the empirical risk, we employ a classical result of Srebro et al. [15] that bounds the generalization gap of linear models in the so-called "low-noise" (i.e., nearly realizable) smooth regime using local Rademacher complexities. Somewhat surprisingly, this simple combination of tools already give sharp risk results that improve upon the state-of-the-art [13, 12, 16] both in terms of tightness of the bounds and the light set of assumptions they rely on.

We remark here that the proof scheme summarized above can be generalized to essentially any gradient method for which one can prove simultaneously bounds on the optimization error and the norm of possible solutions produced by the algorithm. In the sequel, we focus for concreteness on standard gradient descent, but in Appendix B we also give bounds for Stochastic Gradient Descent (SGD), which is shown to admit both of these properties with high probability over the input sample.

An interesting conclusion from our bounds pertains to the significance of early stopping. We see that gradient descent, even when applied on a smooth loss functions that decay rapidly to zero, might overfit with respect the the surrogate loss (i.e., reach a trivial $\Theta(1)$ risk) as the number of steps $T$ grows. The time by which gradient descent starts overfitting depends on the tail decay rate of the loss function: the slower the decay rate, the shorter the time it takes to overfit. Interestingly, a similar phenomenon may not occur for the zero-one loss of the trained predictor. For example, Soudry et al. [14] show that with the logistic loss, gradient descent does not overfit in terms of the zero-one as $T$ approached infinity—whereas, at the same time, it *does* overfit as $T \to \infty$ in terms of the logistic loss itself, as seen from our lower bounds.

**Summary of contributions.**    To summarize, the main contribution of this paper are as follows:

• Our first main result (in Section 3) is a high probability risk upper bound for gradient descent in the setting of separable classification with a convex and smooth loss function. Our bound matches the best known upper bounds [13, 12] and greatly extend them to allow for virtually any convex and smooth loss function, considerably relaxing various assumptions imposed by prior work.

- Our second main result (in Section 4) is a nearly matching lower bound for the risk of gradient descent, establishing the tightness of our analysis given the smoothness and tail-decay conditions. The tightness of our bounds holds across tail decay rates and different regimes of the parameters $T, n$ and $\gamma$. To our knowledge, these constitute the first (let alone tight) risk lower bounds in this context, despite a long line of work on linear classification with separable data.

- We also provide analogous results for Stochastic Gradient Descent with replacement (in Appendix B), mainly to emphasize that that our analysis uses only two elementary properties of the optimization algorithm: low optimization error and low norm of the produced solution. The same analysis can be generalized to any gradient method that admits these two properties.

## 1.2 Additional related work

**Unregularized gradient methods on separable data.** The most relevant work to ours is of Schliserman and Koren [12], who used algorithmic stability and two simple conditions of self-boundedness and realizability which the loss functions hold, to get generalization bounds for gradient methods with constant step size in the general setting of stochastic convex and smooth optimization. Then, they derived risk bounds which hold in expectation for the setting of linear classification with separable data for every loss function which decays to 0. The exact bound was depend in the rate of decaying to 0 of the function. For the Lipschitz case their risk bound with respect to the loss function $\ell$ is $O(\ell^{-1}(\varepsilon)^2/\gamma^2 T + \ell^{-1}(\varepsilon)^2/\gamma^2 n)$ for any choice of $\varepsilon$ such that $\varepsilon/\ell^{-1}(\varepsilon)^2 \leq 1/\gamma^2 T$. For example, for the logistic loss, the bound translates to $O(\log^2(T)/\gamma^2 T + \log^2(T)/\gamma^2 n)$.

In another work, Telgarsky [16], showed a high probability risk bound for $T \leq n$ for Batch Mirror Decent with step size $\eta \simeq 1/\sqrt{T}$ in linear models, using a reference vector, which when selected properly, can be translated to a risk bound of $O(\ell^{-1}(\varepsilon)/\gamma\sqrt{T})$ for gradient descent applied on the loss function $\ell$, and to a $O(\log T/\gamma\sqrt{T})$ for the logistic loss.

**Fast rates for smooth and realizable optimization.** The problem of smooth and realizable optimization, also known as the "low-noise" regime of stochastic optimization, is a very well researched problem. Srebro et al. [15] showed that stochastic gradient descent achieved risk bound of $O(1/n)$ in this setting. For linear models, they also showed that ERM achieve similar fast rates by using local Rademacher complexities. Later [10] showed that SGD converges linearly when the loss function is also strongly convex. In more recent works, [7] used stability arguments to show that SGD with replacement with $T = n$ achieve risk of $O(1/n)$.

**Lower bounds.** A lower bound related to ours appears in Ji and Telgarsky [4]. In this work, the authors showed a lower bound of $\|w'_t - w^*\| \geq \log(n)/\log(T)$. In our work, however, we get lower bound directly for the loss itself and not for this objective. More recently, Shamir [13] showed a lower bound of $\Omega(1/\gamma^2 T)$ for the empirical risk of GD when applied on the logistic loss which is tight up to log factors. When generalizing this technique for other objectives, e.g., functions that decay polynomially to zero, the log factors become polynomial factors and the bound becomes not tight, even in the regime $T \ll n$ where the $1/T$ term in the bounds is dominant. In contrast, we establish nearly tight bounds for virtually any tail decay rate, and in all regimes of $T$ and $n$.

## 2 Problem Setup

We consider the following typical linear classification setting. Let $\mathcal{D}$ be distribution over pairs $(x, y)$, where $x \in \mathbb{R}^d$ is a $d$-dimensional feature vector and $y \in \mathbb{R}$ is a real number that represents the corresponding label. We assume that data is scaled so that $\|x\| \leq 1$ and $|y| \leq 1$ (with probability one with respect to $\mathcal{D}$). We focus on the setting of *separable*, or *realizable*, linear classification with margin. Formally, we make the following assumption.

**Assumption 1** (realizability). There exists a unit vector $w^* \in \mathbb{R}^d$ and $\gamma > 0$ such that $y(w^* \cdot x) \geq \gamma$ almost surely with respect to the distribution $\mathcal{D}$.

Equivalently, we will identify each pair $(x, y)$ with the vector $z = yx$, and realizability implies that $w^* \cdot z \geq \gamma$ with probability 1. Our assumptions then imply that $\|z\| \leq 1$ with probability 1.

Given a nonnegative loss function $\ell : \mathbb{R} \to \mathbb{R}^+$, the objective is to determine a model $w \in \mathbb{R}^d$ that minimizes the (population) risk, defined as the expected value of the loss function over the distribution

$\mathcal{D}$, namely

$$L(w) = \mathbb{E}_{z \sim \mathcal{D}}[\ell(w \cdot z)]. \tag{2}$$

For finding such a model, we use a set of training examples $S = \{z_1, ..., z_n\}$ which drawn i.i.d. from $\mathcal{D}$ and an empirical proxy, the *empirical risk*, which is defined as

$$\widehat{L}(w) = \frac{1}{n} \sum_{i=1}^{n} \ell(w \cdot z_i). \tag{3}$$

### 2.1 Loss functions

The loss functions $\ell$ considered in this paper are nonnegative, convex and $\beta$-smooth.[3] We also require that $\ell$ is strictly monotonically decreasing and $\lim_{u \to \infty} \ell(u) = 0$. The vast majority of loss functions used in supervised learning for classification satisfy these conditions; these include, for example, the logistic loss ($\ell(u) = \log(1 + e^{-u})$), the probit loss ($\ell(u) = -\log(\frac{1}{2} - \frac{1}{2}\operatorname{erf}(u))$), and the squared hinge loss ($\ell(u) = (\max\{1 - u, 0\})^2$).

A main goal of this paper is to quantify how do the achievable bounds on the risk depend on properties of the loss function $\ell$ used, and most crucially on the rate in which $\ell$ decays to zero as its argument approaches infinity. To formalize this, we need a couple of definitions.

**Definition 1** (tail function). We say that $\phi : [0, \infty) \to \mathbb{R}$ is a *tail function* if $\phi$

   (i) is a nonnegative, 1-Lipschitz and $\beta$-smooth convex function;
   (ii) is strictly monotonically decreasing such that $\lim_{u \to \infty} \phi(u) = 0$;
   (iii) satisfies $\phi(0) \geq \frac{1}{2}$ and $|\phi'(0)| \geq \frac{1}{2}$.

Every tail function $\phi$ defines a class of loss functions characterized by the rate $\phi$ decays to zero.

**Definition 2** ($\phi$-tailed class). For a tail function $\phi$, the class $C_{\phi,\beta}$ is the set of all nonnegative, convex, $\beta$-smooth and monotonically decreasing loss functions $\ell : \mathbb{R} \to \mathbb{R}^+$ such that $\ell(u) \leq \phi(u)$ for all $u \geq 0$.

We detail several examples for tail functions in Appendix A.

### 2.2 Gradient Descent

The algorithm that we focus in this paper is standard gradient descent (GD) with a fixed step size $\eta > 0$ applied to the empirical risk $\widehat{L}$; this method is initialized at $w_1 = 0$ and at each step $t = 1, \ldots, T$ performs an update

$$w_{t+1} = w_t - \eta \nabla \widehat{L}(w_t). \tag{4}$$

The algorithm returns the final model, $w_T$.

We remark however that most of the results we present in the sequel can be straightforwardly adapted to other gradient methods; we include results for Stochastic Gradient Descent (SGD) in Appendix B, and the same proof techniques can be used to analyze multi-epoch and/or mini-batched SGD, gradient flow, and more.

## 3 Risk Upper Bounds

We begin by giving a general upper bound for the risk of gradient descent, when the loss function $\ell$ is taken from the class $C_{\phi,\beta}$. Our main result in this section is the following.

**Theorem 1.** *Let $\phi$ be a tail function and let $\ell$ be any loss function from the class $C_{\phi,\beta}$. Fix $T$,$n$ and $\delta > 0$. Then, with probability at least $1 - \delta$ (over the random sample $S$ of size $n$), the output of GD applied on $\widehat{L}$ with step size $\eta \leq 1/2\beta$ initialized at $w_1 = 0$ has*

$$L(w_T) \leq \frac{4K(\phi^{-1}(\varepsilon))^2}{\gamma^2 \eta T} + \frac{32K\beta(\phi^{-1}(\varepsilon))^2 \left(\log^3 n + 4\log\frac{1}{\delta}\right)}{\gamma^2 n} + \frac{4K(\phi^{-1}(\varepsilon))^2 \log\frac{1}{\delta}}{\gamma^2 \eta T n}$$

*for any $\varepsilon \leq \frac{1}{2}$ such that $\eta\gamma^2 T \leq (\phi^{-1}(\varepsilon))^2/\varepsilon$, where $K < 10^5$ is a numeric constant.*

---

[3] A differentiable function $\ell : \mathbb{R} \to \mathbb{R}$ is said to be $\beta$-smooth over $\mathbb{R}$ if $\ell(v) \leq \ell(u) + \ell'(u) \cdot (v - u) + \frac{1}{2}\beta(v - u)^2$ for all $u, v \in \mathbb{R}$.

The expression $(\phi^{-1}(\varepsilon))^2/\varepsilon$ increases indefinitely as $\varepsilon$ approaches 0; therefore, for any $T$, there exists an $\varepsilon > 0$ that satisfies the theorem's condition. For several examples of how this bound is instantiated for different tail decay functions $\phi$, see Table 2 and Appendix A.

In the remainder of this section we prove Theorem 1. The structure of the the proof will be as follows: First, we bound the norm of the GD solution; by smoothness and realizability, we get that the norm will remain small compared to a reference point with small loss value. Second, we get a bound on the optimization error in this setting, the relies on the same reference point. Finally, we use a fundamental result due to Srebro et al. [15] (reviewed in the subsection below) together with both bounds to derive the risk guarantee. As discussed broadly in the introduction, this proof scheme can be generalized to other gradient methods which satisfy the properties of model with low norm and low optimization error.

## 3.1 Preliminaries: Uniform Convergence Using Rademacher Complexity

One property of linear models is that in this class of problems is that we have dimension-independent and algorithm-independent uniform convergence bounds, that enables to bound the difference between the empirical risk and the population risk of a specific model. A main technical tool for bounding this difference is the Rademacher Complexity [2]. The worst-case Rademacher complexity of an hypothesis class $H$ for any sample size $n$ is given by:

$$R_n(H) = \sup_{z_1,\dots z_n} \mathbb{E}_{\sigma \sim \mathrm{Unif}(\{\pm 1\}^n)} \left[ \sup_{h \in H} \frac{1}{n} \left| \sum_{i=1}^n h(z_i)\sigma_i \right| \right].$$

We are interested in models that achieve low empirical risk on smooth objectives. A fundamental result of Srebro et al. [15] bounds the generalization gap under such conditions:

**Proposition 1** (15, Theorem 1). *Let $H$ be a hypothesis class with respect to some non negative and $\beta$-smooth function, $\ell(t \cdot y)$, such that for every $w \in H, x, y, |\ell(wx \cdot y)| \le b$. Then, for any $\delta > 0$ we have, with probability at least $1 - \delta$ over a sample of size $n$, uniformly for all $h \in H$,*

$$L(h) \le \widehat{L}(h) + K\left( \sqrt{\widehat{L}(h)} \left( \sqrt{\beta} \log^{1.5}(n) R_n(H) + \sqrt{\frac{b \log \frac{1}{\delta}}{n}} \right) + \beta \log^3(n) R_n^2(H) + \frac{b \log \frac{1}{\delta}}{n} \right).$$

*where $K < 10^5$ is a numeric constant.*

## 3.2 Properties of Gradient Descent on Smooth Objectives

In this section we prove that GD satisfies the two desired properties- low norm and low optimization error. We begin with showing that the norm of $w_T$, the output of GD after $T$ iterations, is low, as stated in the following lemma,

**Lemma 1.** *Let $\phi$ be a tail function and let $\ell \in C_{\phi,\beta}$. Fix any $\varepsilon > 0$ and a point $w_\varepsilon^* \in \mathbb{R}^d$ such that $\widehat{L}(w_\varepsilon^*) \le \varepsilon$ (exists due to realizability). Then, the output of $T$-steps GD, applied on $\widehat{L}$ with stepsize $\eta \le 1/\beta$ initialized at $w_1 = 0$ has,*

$$\|w_T\| \le 2\|w_\varepsilon^*\| + 2\sqrt{\eta \varepsilon T}.$$

Now, we bound the optimization error of GD on every function $\ell \in C_{\phi,\beta}$, by using a variant of Lemma 13 from [12]. The proof is fairly standard and appears in Appendix C.

**Lemma 2.** *Let $\phi$ be a tail function and let $\ell \in C_{\phi,\beta}$. Fix any $\varepsilon > 0$ and a point $w_\varepsilon^* \in \mathbb{R}^d$ such that $\widehat{L}(w_\varepsilon^*) \le \varepsilon$. Then, the output of $T$-steps GD, applied on $\widehat{L}$ with stepsize $\eta \le 1/\beta$ initialized at $w_1 = 0$ has,*

$$\widehat{L}(w_T) \le \frac{\|w_\varepsilon^*\|^2}{\eta T} + 2\varepsilon.$$

## 3.3 Proof of Theorem 1

We now turn to prove Theorem 1. The proof is a simple consequence of the properties proved above and Proposition 1. We first claim that there exists a model $w_\varepsilon^*$ with low norm such that $\widehat{L}(w_\varepsilon^*) \le \varepsilon$,

which implies, through Lemmas 1 and 2, that $w_T$ of gradient descent has both low optimization error *and* it remains bounded within a ball of small radius. Then, we use Proposition 1 to translate the low optimization error to low risk.

*Proof of Theorem 1.* First, we show that there exists a model $w_\varepsilon^*$ with low norm such that $\widehat{L}(w_\varepsilon^*) \leq \varepsilon$. Let $\varepsilon \leq \frac{1}{2}$ and $\ell \in C_{\phi,\beta}$. By separability, there exists a unit vector $w^*$ such that $w^* \cdot z_i \geq \gamma$ for every $z_i$ in the training set $S$. Moreover, $\ell$ is monotonic decreasing. Then, for $w_\varepsilon^* = (\phi^{-1}(\varepsilon)/\gamma)w^*$ and every $z_i \in S$,

$$\ell(w_\varepsilon^* \cdot z_i) = \ell\left(\frac{\phi^{-1}(\varepsilon)}{\gamma}w^* \cdot z_i\right) \leq \ell\left(\frac{\phi^{-1}(\varepsilon)}{\gamma} \cdot \gamma\right) \leq \ell(\phi^{-1}(\varepsilon)).$$

Then, by the fact that $\phi^{-1}(\varepsilon) \geq 0$, we have $\ell(w_\varepsilon^* \cdot z_i) \leq \ell(\phi^{-1}(\varepsilon)) \leq \phi(\phi^{-1}(\varepsilon)) = \varepsilon$ for all $i$, hence

$$\widehat{L}(w_\varepsilon^*) = \frac{1}{n}\sum_{i=1}^{n}\ell(w_\varepsilon^* \cdot z_i) \leq \varepsilon.$$

Now, for $\varepsilon$ such that $\eta\gamma^2 T \leq (\phi^{-1}(\varepsilon))^2/\varepsilon$, we get by Lemma 1,

$$\|w_T\| \leq 2\|w_\varepsilon^*\| + 2\sqrt{\eta\varepsilon T} \leq \frac{4\phi^{-1}(\varepsilon)}{\gamma}.$$

For the same $\varepsilon$, by Lemma 2,

$$\widehat{L}(w_T) \leq \frac{\|w_\varepsilon^*\|^2}{\eta T} + 2\varepsilon \leq 3\frac{\phi^{-1}(\varepsilon)^2}{\gamma^2\eta T}.$$

Denote $B_\varepsilon = \{w : \|w\| \leq r_\varepsilon\}$, where for brevity $r_\varepsilon = 4\phi^{-1}(\varepsilon)/\gamma$. We have, by Lemma 10, $f(x) \leq 2f(y) + \beta\|x - y\|^2$ for all $x, y \in \mathbb{R}^d$ (see proof in Appendix C). Then, together with the fact that $\|z\|, \|z'\| \leq 1$ and choosing $\varepsilon$ such that $\varepsilon \leq \phi^{-1}(\varepsilon)^2/\gamma^2\eta T$, with probability 1,

$$b = \max_{w \in B_\varepsilon}|\ell(w \cdot z)| \leq 2\ell(w_\varepsilon^* z) + 4\beta r_\varepsilon^2 \leq 2\varepsilon + 4\beta r_\varepsilon^2 \leq \frac{r_\varepsilon^2}{8\eta T} + 4\beta r_\varepsilon^2.$$

Moreover, $B_\varepsilon$ is hypothesis class of linear predictors with norm at most $r_\varepsilon$. We know that the norm of the examples is at most 1, thus, it follows that the Rademacher complexity of $B_\varepsilon$ is $R_n(B_\varepsilon) = r_\varepsilon/\sqrt{n}$ [e.g., 6, Theorem 3].

Now, by the choice of $\varepsilon$, we have $\widehat{L}(w_T) \leq 3r_\varepsilon^2/16\eta T$. Thus, Proposition 1 implies that with probability at least $1 - \delta$, for every $w \in B_\varepsilon$ and any $\varepsilon$ such that $\varepsilon \leq \phi^{-1}(\varepsilon)^2/\gamma^2\eta T$,

$$L(w_T) \leq \frac{3r_\varepsilon^2}{16\eta T} + K\left(\sqrt{\frac{3r_\varepsilon^2}{16\eta T}}\left(\sqrt{\beta}\log^{1.5}n\frac{r_\varepsilon}{\sqrt{n}} + \sqrt{\frac{b\log\frac{1}{\delta}}{n}}\right) + \beta\log^3 n\frac{r_\varepsilon^2}{n} + \frac{b\log\frac{1}{\delta}}{n}\right).$$

Plugging in the bound on $b$, dividing by $r_\varepsilon^2$ and using twice the fact that $xy \leq \frac{1}{2}x^2 + \frac{1}{2}y^2$ for all $x, y$,

$$\frac{L(w_T)}{r_\varepsilon^2} \leq \frac{3}{16\eta T} + K\left(\sqrt{\frac{3}{16\eta T}}\left(\frac{\sqrt{\beta}\log^{1.5}n}{\sqrt{n}} + \sqrt{\frac{(\frac{1}{8\eta T} + 4\beta)\log\frac{1}{\delta}}{n}}\right) + \frac{\beta\log^3 n}{n} + \frac{(\frac{1}{8\eta T} + 4\beta)\log\frac{1}{\delta}}{n}\right)$$

$$\leq \frac{3}{16\eta T} + K\left(\sqrt{\frac{3}{16\eta T}}\left(\frac{\sqrt{\beta}\log^{1.5}n}{\sqrt{n}} + \sqrt{\frac{\log\frac{1}{\delta}}{8\eta Tn} + \frac{4\beta\log\frac{1}{\delta}}{n}}\right) + \frac{\beta\log^3 n}{n} + \frac{\log\frac{1}{\delta}}{8\eta Tn} + \frac{4\beta\log\frac{1}{\delta}}{n}\right)$$

$$\leq \frac{3}{16\eta T} + K\left(\frac{3}{32\eta T} + \frac{1}{2}\left(\frac{\sqrt{\beta}\log^{1.5}n}{\sqrt{n}} + \sqrt{\frac{\log\frac{1}{\delta}}{8\eta Tn} + \frac{4\beta\log\frac{1}{\delta}}{n}}\right)^2 + \frac{\beta\log^3 n}{n} + \frac{\log\frac{1}{\delta}}{8\eta Tn} + \frac{4\beta\log\frac{1}{\delta}}{n}\right)$$

$$\leq \frac{3}{16\eta T} + K\left(\frac{3}{32\eta T} + \frac{\beta\log^3 n}{n} + \frac{\log\frac{1}{\delta}}{8\eta Tn} + \frac{4\beta\log\frac{1}{\delta}}{n} + \frac{\beta\log^3 n}{n} + \frac{\log\frac{1}{\delta}}{8\eta Tn} + \frac{4\beta\log\frac{1}{\delta}}{n}\right)$$

$$\leq \frac{7K}{32\eta T} + \frac{2K\beta\left(\log^3 n + 4\log\frac{1}{\delta}\right)}{n} + \frac{K\log\frac{1}{\delta}}{4\eta Tn}$$

The theorem follows by rearranging the inequality. $\qquad\square$

# 4 Risk Lower Bounds

In this section we present our second main result: a lower bound showing that the bound we proved in Section 3 for gradient descent is essentially tight for loss functions in the class $C_{\phi,\beta}$, for any given tail function $\phi$ and any $\beta > 0$. Formally, we prove the following theorem.

**Theorem 2.** *There exists a constant $C$ such that the following holds. For any tail function $\phi$, sample size $n \geq 35$ and any $T$, there exist a distribution $\mathcal{D}$ and a loss function $\ell \in C_{\phi,\beta}$, such that for $T$-steps GD over a sample $S = \{z_i\}_{i=1}^n$ sampled i.i.d. from $\mathcal{D}$, initialized at $w_1 = 0$ with stepsize $\eta \leq 1/2\beta$, it holds that*

$$\mathbb{E}[L(w_T)] \geq C \frac{\beta(\phi^{-1}(128\varepsilon))^2}{\gamma^2 n} + C \frac{(\phi^{-1}(8\varepsilon))^2}{\gamma^2 \eta T}.$$

*for any $\varepsilon < \frac{1}{256}$ such that $\eta\gamma^2 T \geq (\phi^{-1}(\varepsilon))^2 / \varepsilon$.*

We remark that the right-hand side of the bound is well defined, as we restrict $\varepsilon$ to be sufficiently small so as to ensure that all arguments to $\phi^{-1}$ are at most $\frac{1}{2}$ (recall that $\phi$ admits all values in $[0, \frac{1}{2}]$ due to our assumptions that $\phi(0) \geq \frac{1}{2}$). Further, the lower bound above matches the upper bound given in Theorem 1 up to constants, unless the tail function $\phi$ decays extremely slowly, and slower than any polynomial (at this point, however, the entire bound becomes almost vacuous).

To prove Theorem 2, we consider two different regimes: the first is where $T \gg n$, when the first term in the right-hand side of the bound is dominant; and the $T \ll n$ regime where the second term is dominant. We begin by focusing on the first regime, and prove the following.

**Lemma 3.** *There exists a constant $C_1$ such that the following holds. For any tail function $\phi$, sample size $n \geq 35$ and any $\gamma$ and $T$, there exist a distribution $\mathcal{D}$ with margin $\gamma$, a loss function $\ell \in C_{\phi,\beta}$ such that for GD over a sample $S = \{z_i\}_{i=1}^n$ sampled i.i.d. from $\mathcal{D}$, initialized at $w_1 = 0$ with stepsize $\eta \leq 1/2\beta$, it holds that*

$$\mathbb{E}[L(w_T)] \geq C_1 \frac{\beta\phi^{-1}(128\varepsilon)^2}{\gamma^2 n},$$

*for any $\varepsilon < \frac{1}{256}$ such that $\eta\gamma^2 T \geq (\phi^{-1}(\varepsilon))^2 / \varepsilon$.*

To prove the lemma, we construct a learning problem for which the risk of GD can be lower bounded. We design a loss function $\ell$ which exhibits a decay to zero at a rate similar to that of the function $\phi$ for $x > 0$, and behaves as a quadratic function for $x \leq 0$. The distribution $\mathcal{D}$ is constructed such that there are two fixed examples in its support, denoted $z_1$ and $z_2$, that are both sampled with constant probability and are almost opposite in direction to each other: $z_2$ is aligned with $-z_1$ except for a small component, denoted $v$, which is orthogonal to $z_1$. Crucially, since $z_1$ and $z_2$ point in nearly opposite directions, minimizing optimization error requires the GD iterate to have a substantial component aligned with the vector $v$ (otherwise, the loss for $z_2$ will be large). Finally, we define another example, denoted as $z_3$, that has a significant component in the opposite direction $-v$. We arrange the distribution $\mathcal{D}$ so that $z_3$ is sampled with with probability roughly $1/n$. The lower bound now follows by observing that, if $z_3$ is sampled at test time, but did not appear in the training set (this happens with probability $\approx 1/n$), the test error is quadratically large in the magnitude of the GD iterate along $-v$.

We now turn to formally proving Lemma 3.

*Proof.* Given $\gamma \leq \frac{1}{8}$, let us define the following distribution $\mathcal{D}$:

$$\mathcal{D} = \begin{cases} z_1 := (1, 0, 0) & \text{with prob. } \frac{59}{64}(1 - \frac{1}{n}); \\ z_2 := (-\frac{1}{2}, 3\gamma, 0) & \text{with prob. } \frac{5}{64}(1 - \frac{1}{n}); \\ z_3 := (0, -\frac{1}{8}, 4\gamma + \frac{1}{4}) & \text{with prob. } \frac{1}{n}, \end{cases}$$

and loss function:

$$\ell(x) = \begin{cases} \phi(x) & x \geq 0; \\ \phi(0) + \phi'(0)x + \frac{\beta}{2}x^2 & x < 0. \end{cases}$$

First, note that the distribution is separable: for $w^* = (\gamma, \frac{1}{2}, \frac{1}{4})$ it holds that $w^* z_i = \gamma$ for every $i \in \{1, 2, 3\}$. Moreover, Lemma 11 in Appendix D ensures that indeed $\ell \in C_{\phi,\beta}$.

Next, let $S$ be a sample of $n$ i.i.d. examples from $\mathcal{D}$ and let $z' \sim \mathcal{D}$ be a validation example independent from $S$. Denote by $\delta_2 \in [0, 1]$ the fraction of appearances of $z_2$ in the sample $S$, and by $A_1, A_2$ the following events;

$$A_1 = \{z' = z_3 \wedge z_3 \notin S\}, \qquad A_2 = \left\{\delta_2 \in \left[\tfrac{1}{32}, \tfrac{1}{8}\right]\right\}.$$

In Lemma 12 (found in Appendix D), we show that

$$\Pr(A_1 \cap A_2) \geq \frac{1}{120en}. \tag{5}$$

Furthermore, as in the proof of Theorem 1, there exists a vector $w_\varepsilon^*$ which holds $\|w_\varepsilon^*\| \leq \phi^{-1}(\varepsilon)/\gamma$. Then by Lemma 2 and the choice of $\varepsilon$,

$$\widehat{L}(w_T) \leq 2\varepsilon + \frac{2\phi^{-1}(\varepsilon)^2}{\gamma^2} \leq 4\varepsilon. \tag{6}$$

For the remainder of the proof, we condition on the event $A_1 \cap A_2$. First, we show that $w_t \cdot z_2 \geq 0$. Indeed, if it were not the case, then $\ell(w_T \cdot z_2) > \phi(0)$; together with Eq. (6) we obtain

$$\frac{1}{64} > 4\varepsilon \geq \widehat{L}(w_T) \geq \delta_2 \ell(w_T \cdot z_2) \geq \frac{1}{32}\phi(0).$$

which is a contradiction to $\phi(0) \geq \frac{1}{2}$. Moreover, $w_T(1) \geq 0$. Again, we show this by contradiction. Conditioned on $A_2$, we have $\delta_1 > \frac{7}{8}$. Then, if $w_T(1) < 0$, $\ell(w_T \cdot z_1) > \phi(0)$, and

$$\frac{1}{64} > 4\varepsilon \geq \widehat{L}(w_T) \geq \delta_1 \ell(w_T \cdot z_1) > \frac{7}{8}\phi(0) \geq \frac{7}{16}.$$

which is a contradiction. In addition, we notice that $z_3$ is the only possible example whose third entry is non zero. Given the event $A_1$, we know that $z_3$ is not in $S$. Equivalently, for every $z \in S$, $z(3) = 0$. As a result,

$$w_t(3) = -\eta \sum_{s=1}^{t-1} \nabla \widehat{L}(w_t) = -\eta \sum_{s=1}^{t-1} \frac{1}{n} \sum_{z \in S} z(3)\ell'(w_s z) = 0.$$

Then, we get that,

$$w_T \cdot z_3 = -\frac{1}{8}w_t(2). \tag{7}$$

Then, using the fact that $w_T \cdot z_2 \geq 0$, $\ell(w_T \cdot z_2) = \phi(w_T \cdot z_2)$, and conditioned on $A_2$, we have

$$\ell(w_T \cdot z_2) = \phi(w_T \cdot z_2) \leq 32\widehat{L}(w_T)),$$

which implies

$$w_T \cdot z_2 \geq \phi^{-1}(32\widehat{L}(w_T)). \tag{8}$$

Therefore, by combining Eq. (8) with the fact that $w_t(1) \geq 0$,

$$3\gamma w_T(2) \geq -\frac{w_T(1)}{2} + 3\gamma w_T(2) = w_T \cdot z_2 \geq \phi^{-1}(32\widehat{L}(w_T)).$$

which implies, $w_T(2) \geq \frac{1}{3\gamma}\phi^{-1}(32\widehat{L}(w_T))$. By Eq. (7),

$$w_T \cdot z_3 = -\frac{1}{8}w_T(2) \leq -\frac{1}{24\gamma}\phi^{-1}(32\widehat{L}(w_T)).$$

We therefore see that for every $\varepsilon$ such that $\varepsilon \geq (\phi^{-1}(\varepsilon))^2/\gamma^2 T\eta$,

$$\ell(w_T \cdot z_3) \geq \frac{\beta}{2}(w_T \cdot z_3)^2 \geq \frac{\beta}{2}\left(\frac{1}{24\gamma}\phi^{-1}(32\widehat{L}(w_T))\right)^2 \geq \frac{\beta}{1152\gamma^2}\left(\phi^{-1}(128\varepsilon)\right)^2,$$

where in the final inequality we again used Eq. (6). We conclude the proof using Eq. (5) and the law of total expectation,

$$\mathbb{E}[L(w_T)] = \mathbb{E}[\ell(w_T \cdot z')] \geq \mathbb{E}[\ell(w_T \cdot z') \mid A_1 \cap A_2]\Pr(A_1 \cap A_2).$$

(Expectations here are taken with respect to both the sample $S$ and the validation example $z'$.) $\qquad \square$

Second, we focus the second expression in the lower bound of Theorem 2, which is dominant in the early stages of optimization.

**Lemma 4.** *There exists a constant $C_2$ such that the following holds. For any tail function $\phi$, and for any $n, T$ and $\gamma$, there exist a distribution $\mathcal{D}$ with margin $\gamma$, a loss function $\ell \in C_{\phi,\beta}$ such that for GD initialized at $w_1 = 0$ with stepsize $\eta \leq 1/2\beta$ over an i.i.d. sample $S = \{z_i\}_{i=1}^n$ from $\mathcal{D}$ and $w_1 = 0$ holds*

$$\mathbb{E}[L(w_T)] \geq C_2 \frac{(\phi^{-1}(8\varepsilon))^2}{\gamma^2 T \eta},$$

*for any $\varepsilon \leq \frac{1}{16}$ such that $\eta\gamma^2 T \geq \phi^{-1}(\varepsilon)^2/\varepsilon$.*

The argument for proving Lemma 4 is similar to that of Lemma 3. Here we define a 1-Lipschitz loss function $\ell$ that decays to zero at the same rate as $\phi$, and a distribution such that there is a possible example $z_1$ which is sampled with high probability and an almost "opposite" example $z_2$, that with constant probability, appears rarely in the training set $S$. The lower bound follows from the fact that although $z_2$ appears in the dataset, the gradients of the loss function are bounded and thus not large enough so as to make the trained predictor classify $z_2$ correctly. The full proof can be found in Appendix D.2.

Finally, we derive Theorem 2 directly from Lemmas 3 and 4:

*Proof of Theorem 2.* Let $C = \frac{1}{2}\min\{C_1, C_2\}$, where $C_1$ and $C_2$ are the constants from Lemmas 3 and 4, respectively. If $(\phi^{-1}(8\varepsilon))^2/\gamma^2 T \eta \geq \beta(\phi^{-1}(128\varepsilon))^2/\gamma^2 n$, the theorem follows from Lemma 4; otherwise, it follows from Lemma 3. $\qquad\square$

### Acknowledgments

Funded by the European Union (ERC, OPTGEN, 101078075). Views and opinions expressed are however those of the author(s) only and do not necessarily reflect those of the European Union or the European Research Council. Neither the European Union nor the granting authority can be held responsible for them. This work received additional support from the Israel Science Foundation (grant number 2549/19), from the Len Blavatnik and the Blavatnik Family foundation, and from the Adelis Foundation.

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
