# A  Derivation of Risk Bounds in Table 2

In this section, we explain how to apply our risk bounds for GD with several popular choices of $\phi$-tailed classes.

## A.1  Exponentially-tailed functions

We begin with functions that decay exponentially to zero. For any $\alpha > 0$ we define following the tail function, $\phi_\alpha(y) = e^{-y^\alpha}$. There are a lot of popular loss functions loss function $\ell \in C_{\phi_\alpha,\beta}$, e.g., the logistic loss $\ell(y) = \log(1 + e^{-y})$. holds $\ell \in C_{\phi_1,1}$. For this class of functions function, we get the following risk bound:

**Corollary 1.** *If $\ell \in C_{\phi_\alpha,\beta}$, the output of gradient descent on $\widehat{F}$ with step size $\eta = \frac{1}{2\beta}$ and $w_1 = 0$ holds, with probability of $1 - \delta$, for a suitable choice of $\varepsilon$,*

$$L(w_T) = \tilde{O}\left(\frac{\log^{\frac{2}{\alpha}} T}{\gamma^2 T} + \frac{\log^{\frac{2}{\alpha}} T}{\gamma^2 n}\right).$$

*Moreover, if $\gamma^2 T \geq 256^4$ and $\frac{\alpha}{4} \geq \frac{\log\log(\eta\gamma^2 T)}{\log(\eta\gamma^2 T)}$, there exists a function $\bar{\ell} \in C_{\phi_\alpha,\beta}$ and a distribution $\mathcal{D}$ such that*

$$\mathbb{E}[\bar{L}(w_T)] = \tilde{\Omega}\left(\frac{\log^{\frac{2}{\alpha}} T}{\gamma^2 T} + \frac{\log^{\frac{2}{\alpha}} T}{\gamma^2 n}\right).$$

*Proof.* First, $\phi^{-1}(x) = \log^{\frac{1}{\alpha}}(\frac{1}{x})$. For the upper bound we choose $\varepsilon = 1/\eta\gamma^2 T$, which holds $\eta\gamma^2 T \leq (\phi^{-1}(\varepsilon))^2/\varepsilon$. Then,

$$\phi^{-1}(\varepsilon) = \log^{\frac{1}{\alpha}}(\gamma^2\eta T)$$

Then, by Theorem 1, with probability $1 - \delta$,

$$\bar{L}(w_T) \leq \frac{4K(\phi^{-1}(\varepsilon))^2}{\gamma^2\eta T} + \frac{32K\beta(\phi^{-1}(\varepsilon))^2\left(\log^3 n + 4\log\frac{1}{\delta}\right)}{\gamma^2 n} + \frac{4K(\phi^{-1}(\varepsilon))^2\log\frac{1}{\delta}}{\gamma^2\eta T n}$$

$$= O\left(\frac{(\log\gamma^2 T)^{\frac{2}{\alpha}}}{\gamma^2 T} + \frac{(\log\gamma^2 T)^{\frac{2}{\alpha}}\left(\log^3 n + 4\log\frac{1}{\delta}\right)}{\gamma^2 n}\right)$$

For the lower bound we choose, $\varepsilon = (\gamma^2\eta T)^{-\frac{1}{2}}$, which holds

$$(\eta\gamma^2 T\varepsilon)^{\frac{\alpha}{2}} = (\eta\gamma^2 T)^{\frac{\alpha}{2}}\varepsilon^{\frac{\alpha}{2}} = (\eta\gamma^2 T)^{\frac{\alpha}{2}} \cdot (\eta\gamma^2 T)^{\frac{-\alpha}{4}} = (\eta\gamma^2 T)^{\frac{\alpha}{4}} \geq (\eta\gamma^2 T)^{\frac{\log\log(\eta\gamma^2 T)}{\log(\eta\gamma^2 T)}} = \log(\eta\gamma^2 T)$$

$$\eta\gamma^2 T\varepsilon \geq \log(\eta\gamma^2 T)^{\frac{2}{\alpha}} = \log^{\frac{2}{\alpha}}\frac{1}{\varepsilon^2} = 2^{\frac{2}{\alpha}}\log^{\frac{2}{\alpha}}(\frac{1}{\varepsilon}) \geq \log^{\frac{2}{\alpha}}(\frac{1}{\varepsilon})$$

Then,

$$\phi^{-1}(8\varepsilon) = \log^{\frac{1}{\alpha}}(\frac{1}{8}\gamma\sqrt{\eta T}) = (\frac{1}{2})^{\frac{1}{\alpha}}\log^{\frac{1}{\alpha}}(\frac{1}{64}\gamma^2\eta T) \geq (\frac{1}{2})^{\frac{1}{\alpha}}\log^{\frac{1}{\alpha}}(4\gamma\sqrt{\eta T}) \geq (\frac{1}{2})^{\frac{2}{\alpha}}\log^{\frac{1}{\alpha}}(\gamma^2\eta T)$$

$$\phi^{-1}(128\varepsilon) = \log^{\frac{1}{\alpha}}(\frac{1}{128}\gamma\sqrt{\eta T}) = (\frac{1}{2})^{\frac{1}{\alpha}}\log^{\frac{1}{\alpha}}(\frac{1}{128^2}\gamma^2\eta T) \geq (\frac{1}{2})^{\frac{1}{\alpha}}\log^{\frac{1}{\alpha}}(4\gamma\sqrt{\eta T}) \geq (\frac{1}{2})^{\frac{2}{\alpha}}\log^{\frac{1}{\alpha}}(\gamma^2\eta T)$$

As a result, by Theorem 2, there exists a function $\bar{\ell} \in C_{\phi_\alpha,\beta}$ and a distribution

$$\mathbb{E}[\bar{L}(w_T)] \geq C\frac{\beta(\phi^{-1}(128\varepsilon))^2}{\gamma^2 n} + C\frac{(\phi^{-1}(8\varepsilon))^2}{\gamma^2\eta T}$$

$$= \Omega\left(\frac{\log^{\frac{2}{\alpha}}(\gamma^2 T)}{\gamma^2 n} + \frac{\log^{\frac{2}{\alpha}}(\gamma^2 T)}{\gamma^2 T}\right).$$

$\square$

## A.2  Polynomially-tailed losses

Now we turn to discuss loss functions with polynomially-decaying tails. For any $\alpha > 0$ we define following the tail function, $\phi_\alpha(y) = y^{-\alpha}$. Again, we get risk bounds for every loss function $\ell \in C_{\phi_\alpha,\beta}$.

**Corollary 2.** *If $\ell \in C_{\phi_\alpha,\beta}$, the output of gradient descent on $\widehat{F}$ with step size $\eta = \frac{1}{2\beta}$ and $w_1 = 0$ holds, with probability of $1 - \delta$, for a suitable choice of $\varepsilon$,*

$$L(w_T) = \tilde{O}\left(\left(\frac{1}{\gamma}\right)^{\frac{2\alpha}{2+\alpha}}\left(\frac{1}{T^{\frac{\alpha}{2+\alpha}}} + \frac{T^{\frac{2}{\alpha+2}}}{n}\right)\right).$$

*Moreover, if $\gamma^2 T \geq$ there exists a function $\bar{\ell} \in C_{\phi_\alpha,\beta}$ and a distribution $\mathcal{D}$ such that*

$$\mathbb{E}[\bar{L}(w_T)] = \Omega\left(\left(\frac{1}{\gamma}\right)^{\frac{2\alpha}{2+\alpha}}\left(\frac{1}{T^{\frac{\alpha}{2+\alpha}}} + \frac{T^{\frac{2}{\alpha+2}}}{n}\right)\right).$$

*Proof.* First, we note that $\phi^{-1}(x) = x^{-\frac{1}{\alpha}}$. Then, for both upper and lower bound, we choose $\varepsilon = \left(\frac{1}{\eta\gamma^2 T}\right)^{\frac{\alpha}{2+\alpha}}$ which holds $(\phi^{-1}(\varepsilon))^2/\varepsilon = \eta\gamma^2 T$. Now,

$$\phi^{-1}(\varepsilon) = (\eta\gamma^2 T)^{\frac{1}{\alpha+2}}$$

Then, by Theorem 1, with probability $1 - \delta$,

$$
\begin{aligned}
\bar{L}(w_T) &\leq \frac{4K(\phi^{-1}(\varepsilon))^2}{\gamma^2\eta T} + \frac{32K\beta(\phi^{-1}(\varepsilon))^2\left(\log^3 n + 4\log\frac{1}{\delta}\right)}{\gamma^2 n} + \frac{4K(\phi^{-1}(\varepsilon))^2\log\frac{1}{\delta}}{\gamma^2\eta Tn} \\
&= O\left(\frac{(\gamma^2 T)^{\frac{2}{\alpha+2}}}{\gamma^2 T} + \frac{((\gamma^2 T)^{\frac{2}{\alpha+2}}\left(\log^3 n + 4\log\frac{1}{\delta}\right))}{\gamma^2 n}\right) \\
&= O\left(\frac{1}{(\gamma^2 T)^{\frac{\alpha}{\alpha+2}}} + \frac{T^{\frac{2}{\alpha+2}}\left(\log^3 n + 4\log\frac{1}{\delta}\right)}{\gamma^{\frac{\alpha}{\alpha+2}} n}\right)
\end{aligned}
$$

Moreover,

$$\phi^{-1}(8\varepsilon) = 8^{\frac{-1}{\alpha}}(\eta\gamma^2 T)^{\frac{1}{\alpha+2}}$$

$$\phi^{-1}(128\varepsilon) = 128^{\frac{-1}{\alpha}}(\eta\gamma^2 T)^{\frac{1}{\alpha+2}}$$

As a result, by Theorem 2, there exists a function $\bar{\ell} \in C_{\phi_\alpha,\beta}$ and a distribution

$$
\begin{aligned}
\mathbb{E}[\bar{L}(w_T)] &\geq C\frac{\beta(\phi^{-1}(128\varepsilon))^2}{\gamma^2 n} + C\frac{(\phi^{-1}(8\varepsilon))^2}{\gamma^2\eta T} \\
&= \Omega\left(\frac{(\gamma^2 T)^{\frac{2}{\alpha+2}}}{\gamma^2 n} + \frac{(\gamma^2 T)^{\frac{2}{\alpha+2}}}{\gamma^2 T}\right). \\
&= \Omega\left(\frac{1}{(\gamma^2 T)^{\frac{\alpha}{\alpha+2}}} + \frac{T^{\frac{2}{\alpha+2}}}{\gamma^{\frac{\alpha}{\alpha+2}} n}\right)
\end{aligned}
$$

$\square$

## B  Upper bound for Stochastic Gradient Descent With Replacement

We also show generalization bound for Stochastic Gradient Descent (SGD) which is a randomized algorithm which achieves low optimization error in high probability.

Given a dataset $S = \{z_1, ...z_n\}$, we define $\ell_i(w) = \ell(w \cdot z_i)$. Then, SGD with replacement is initialized at a point $w_1 = 0$ and at each step $t = 1, \ldots, T$, samples randomly an index $i_t \in [n]$ and performs an update

$$w_{t+1} = w_t - \eta\nabla\ell_{i_t}(w_t), \tag{9}$$

where $\eta > 0$ is the step size of the algorithm. We consider a standard variant of SGD that returns the average iterate, namely $\overline{w}_T = \frac{1}{T} \sum_{t=1}^{T} w_t$. We start with showing a deterministic bound on the norm of the iterate of the algorithm. Then, we show high probability bound for the empirical risk $\widehat{L}$. Then, the proof of the risk bound is identical to the proof of the risk bound for GD (see Theorem 1). As a result, we will not show the full proof, but only the bounds on the norm (see Lemma 5) and the empirical risk (see Lemma 8).

**Lemma 5.** *Let $T$. Let $\phi$ be a tail function and let $\ell \in C_{\phi,\beta}$. Assume that for every $\varepsilon > 0$ there exists a point $w_\varepsilon^*$ such that for every $i$, $\ell_i(w_\varepsilon^*) \leq \varepsilon$. Then, the output of SGD with replacement, applied on $\widehat{L}$ with step size $\eta \leq \frac{1}{\beta}$ initialized at $w_1 = 0$ has,*

$$\|\overline{w}_T\| \leq 2\|w_\varepsilon^*\| + 2\sqrt{\eta \varepsilon T}.$$

*Proof.* By Lemma 9 (see Appendix C), we know that for every $w$, $\|\nabla \ell_{i_t}(w)\|^2 \leq 2\beta \ell_{i_t}(w)$. Therefore, by using $\eta \leq \frac{1}{\beta}$, for every $\varepsilon$,

$$
\begin{aligned}
\|w_{t+1} - w_\varepsilon^*\|^2 &= \|w_t - \eta \nabla \ell_{i_t}(w_t) - w_\varepsilon^*\|^2 \\
&= \|w_t - w_\varepsilon^*\|^2 - 2\eta \langle w_t - w_\varepsilon^*, \nabla \ell_{i_t} \rangle + \eta^2 \|\nabla \ell_{i_t}(w_t)\|^2 \\
&\leq \|w_t - w_\varepsilon^*\|^2 + 2\eta \ell_{i_t}(w_\varepsilon^*) - 2\eta \ell_{i_t}(w_\varepsilon^*) + 2\beta \eta^2 \ell_{i_t}(w_t) \\
&\leq \|w_t - w_\varepsilon^*\|^2 + 2\eta \ell_{i_t}(w_\varepsilon^*) \\
&\leq \|w_t - w_\varepsilon^*\|^2 + 2\eta \varepsilon.
\end{aligned}
$$

By summing until time $T$,

$$
\begin{aligned}
\|w_T - w_\varepsilon^*\|^2 &\leq \|w_1 - w_\varepsilon^*\|^2 + 2T\eta\varepsilon \\
&= \|w_\varepsilon^*\|^2 + 2\eta\varepsilon T.
\end{aligned}
$$

By taking a square root, using the fact that $\forall x, y \geq 0 \ \sqrt{x+y} \leq \sqrt{x} + \sqrt{y}$,

$$\|w_T - w_\varepsilon^*\| \leq \|w_\varepsilon^*\| + 2\sqrt{\eta\varepsilon T}. \tag{10}$$

and by using triangle inequality,

$$\|w_T\| \leq \|w_T - w_\varepsilon^*\| + \|w_\varepsilon^*\| \leq 2\|w_\varepsilon^*\| + 2\sqrt{\eta\varepsilon T}.$$

and finally by another use of traingle inequality,

$$\|\overline{w}_T\| \leq \frac{1}{T} \sum_{t=1}^{T} \|w_T\| \leq 2\|w_\varepsilon^*\| + 2\sqrt{\eta\varepsilon T}.$$

$\square$

Now, we prove the next lemma and bound the regret of $SGD$ with replacement.

**Lemma 6.** *Let $T$. Let $\phi$ be a tail function and let $\ell \in C_{\phi,\beta}$. Then, the iterate of SGD with replacement, applied on $\widehat{L}$ with step size $\eta \leq \frac{1}{2\beta}$ initialized at $w_1 = 0$ has, for every $w \in R^d$,*

$$\frac{1}{T} \sum_{t=1}^{T} \ell(w_t \cdot z_{i_t}) - \frac{2}{T} \sum_{t=1}^{T} \ell(w \cdot z_{i_t}) \leq \frac{\|w\|^2}{2\eta T}.$$

*Proof.* The proof is almost the same as of [12]. For every $w$, iteration $j$ and possible $i_j$, by Lemma 9 and convexity,

$$
\begin{aligned}
\|w_{j+1} - w\|^2 &\leq \|w_j - w\|^2 - 2\eta \langle \nabla \ell_{i_j}(w_j)(w_j - w) \rangle + \eta^2 \|\nabla \ell_{i_j}(w_j)\|^2 \\
&\leq \|w_j - w\|^2 + 2\eta \ell(w \cdot z_{i_j}) - 2\eta \ell(w_j \cdot z_{i_j}) + 2\eta^2 L \ell(w_j \cdot z_{i_j}) \\
&\leq \|w_j - w\|^2 + 2\eta \ell(w \cdot z_{i_j}) - \eta \ell(w_j \cdot z_{i_j}).
\end{aligned}
$$

Taking average on $j = 1...T$, we get

$$\frac{1}{T} \sum_{t=1}^{T} \ell(w_t \cdot z_{i_t}) - \frac{2}{T} \sum_{t=1}^{T} \ell(w \cdot z_{i_t}) \leq \frac{\|w\|^2}{2\eta T}.$$

$\square$

We use the following concentration bound, taken from [1], Lemma 9.

**Lemma 7.** *Suppose $Z_1, \dots Z_T$ is a sequence such that for every $t \leq T$, $\mathbb{E}\,(Z_t | Z_1 \dots Z_{t-1}) = 0$. and let $Z = \sum_{t=1}^{T} Z_t$. Assume that $|Z_t| \leq b$ for all $t$, and define $V = \sum_{t=1}^{T} \mathbb{E}\left[Z_t^2 \mid \mathcal{F}_{t-1}\right]$. Then, for any $\delta > 0$ and $\lambda \in [0, 1/b]$, with probability of $1 - \delta$,*

$$Z \leq \lambda V + \frac{1}{\lambda} \log \frac{1}{\delta}.$$

Then, we can get high probability guarantee for SGD with replacement.

**Lemma 8.** *Let $T$ and $\delta > 0$. Let $\phi$ be a tail function and let $\ell \in C_{\phi, \beta}$. Assume that for every $\varepsilon > 0$ there exists a point $w_\varepsilon^*$ such such that for every $i$, $\ell_i(w_\varepsilon^*) \leq \varepsilon$. Then, the output of SGD with replacement, applied on $\widehat{L}$ with step size $\eta \leq \frac{1}{\beta}$ initialized at $w_1 = 0$ has, with probability of $1 - \delta$,*

$$\widehat{L}\,(\bar{w}_T) \leq \frac{\|w_\varepsilon^*\|^2}{\eta T} + 5\varepsilon + \frac{8\left(3\varepsilon + 16\beta\|w_\varepsilon^*\|^2 + 16\eta\varepsilon T\right)}{T} \log\left(\frac{1}{\delta}\right).$$

*Proof.* Let $w$. We define

$$Z_t = \ell(w_\varepsilon^* \cdot z_{i_t}) - \widehat{L}(w_\varepsilon^*) - \ell(w_t \cdot z_{i_t}) + \widehat{L}(w_t).$$

By the fact that $w_t, i_t$ is independent, $\mathbb{E}\,(Z_t | Z_1 \dots Z_{t-1}) = 0$. First,

$$\begin{aligned}
b &= \max_t |Z_t| \\
&\leq \varepsilon + \max_{t,i} \ell(w_t \cdot z_i) && (\ell(w_\varepsilon^* \cdot z_{i_t}) \leq \varepsilon, \text{ definition of } \widehat{L}, \text{ nonnegativity}) \\
&\leq \varepsilon + \max_{i, \|w\| \leq 2\|w_\varepsilon^*\| + 2\sqrt{\eta\varepsilon T}} \ell(w \cdot z_i) && (\text{Lemma 5}) \\
&\leq \varepsilon + 2\max_i \ell(w_\varepsilon^* \cdot z_i) + 16\beta\|w_\varepsilon^*\|^2 + 16\eta\varepsilon T && (\text{Lemma 10}) \\
&\leq 3\varepsilon + 16\beta\|w_\varepsilon^*\|^2 + 16\eta\varepsilon T.
\end{aligned}$$

We denote $b = 3\varepsilon + 16\beta\|w_\varepsilon^*\|^2 + 16\eta\varepsilon T$. Then,

$$\max_t |Z_t| \leq 3\varepsilon + 16\beta\|w_\varepsilon^*\|^2 + 16\eta\varepsilon T = b.$$

Moreover, we denote $\mathbb{E}_t[\cdot]$ the expectation conditioned on the randomness of the algorithm before step $t$. Then,

$$\begin{aligned}
V &= \sum_{t=1}^{T} \mathbb{E}_t\left[\left(\ell(w_t \cdot z_{i_t}) - \widehat{L}(w_t) - \ell(w \cdot z_{i_t}) + \widehat{L}(w)\right)^2\right] \\
&\leq b \sum_{i=1}^{T} \mathbb{E}_t\left[\left|\ell(w_t \cdot z_{i_t}) - \widehat{L}(w_t) - \ell(w \cdot z_{i_t}) + \widehat{L}(w)\right|\right] \\
&\leq b \sum_{i=1}^{T} \mathbb{E}_t\left[\ell(w_t \cdot z_{i_t})|\right] + b \sum_{i=1}^{T} \mathbb{E}_t\left[\ell(w \cdot z_{i_t})|\right] + b \sum_{i=1}^{T} \mathbb{E}_t\left[\widehat{L}(w_t)|\right] + b \sum_{i=1}^{T} \mathbb{E}_t\left[\widehat{L}(w)|\right] \\
&\leq 2b \sum_{i=1}^{T} \widehat{L}(w_t) + 2bT \sum_{i=1}^{T} \widehat{L}(w)
\end{aligned}$$

By Lemma 7, with probability $1 - \delta$ by choosing $\lambda = \frac{1}{4b}$,

$$\sum_{t=1}^{T} Z_t \leq \frac{1}{2} \sum_{i=1}^{T} \widehat{L}(w_t) + \frac{1}{2}T\widehat{L}(w) + \frac{4b}{T} \log\left(\frac{1}{\delta}\right).$$

Then, with probability $1 - \delta$,

$$\begin{aligned}
\frac{1}{T} \sum_{i=1}^{T} \widehat{L}(w_t) - \widehat{L}(w) &= \frac{1}{T} \sum_{t=1}^{T} \ell(w_t \cdot z_{i_t}) - \ell(w \cdot z_{i_t}) + \frac{1}{T} \sum_{t=1}^{T} Z_t \\
&\leq \frac{1}{T} \sum_{i=1}^{T} \ell(w_t \cdot z_{i_t}) - \frac{1}{T} \sum_{i=1}^{T} \ell(w \cdot z_{i_t}) + \frac{1}{2T} \sum_{t=1}^{T} \widehat{L}(w_t) + \frac{1}{2}\widehat{L}(w) + 4b \log\left(\frac{1}{\delta}\right).
\end{aligned}$$

By organizing,

$$\frac{1}{2T}\sum_{i=1}^{T}\widehat{L}(w_t) \leq \frac{1}{T}\sum_{i=1}^{T}\ell(w_t \cdot z_{i_t}) - \frac{1}{T}\sum_{i=1}^{T}\ell(w \cdot z_{i_t}) + \frac{3}{2}\widehat{L}(w) + \frac{4b}{T}\log\left(\frac{1}{\delta}\right).$$

Moreover, with probability $1 - \delta$, by Lemma 6 and Jensen inequality

$$\widehat{L}\left(\frac{1}{T}\sum_{i=1}^{T}w_t\right) \leq \frac{1}{T}\sum_{i=1}^{T}\widehat{L}(w_t) \leq \frac{2}{T}\sum_{i=1}^{T}\ell(w_t \cdot z_{i_t}) - \frac{2}{T}\sum_{i=1}^{T}\ell(w \cdot z_{i_t}) + 3\widehat{L}(w) + \frac{8b}{T}\log\left(\frac{1}{\delta}\right)$$

$$= 2\left(\frac{1}{T}\sum_{i=1}^{T}\ell(w_t \cdot z_{i_t}) - \frac{2}{T}\sum_{i=1}^{T}\ell(w \cdot z_{i_t})\right) + 2\widehat{L}(w) + 3\widehat{L}(w) + \frac{8b}{T}\log\left(\frac{1}{\delta}\right)$$

$$\leq \frac{\|w\|^2}{\eta T} + 5\widehat{L}(w) + \frac{8b}{T}\log\left(\frac{1}{\delta}\right).$$

Finally, for $w = w_\varepsilon^*$,

$$\widehat{L}(\bar{w}_T) \leq \frac{\|w_\varepsilon^*\|^2}{\eta T} + 5\varepsilon + \frac{8\left(3\varepsilon + 16\beta\|w_\varepsilon^*\|^2 + 16\eta\varepsilon T\right)}{T}\log\left(\frac{1}{\delta}\right).$$

$\square$

## C   Proofs of Section 3

*Proof of Lemma 1.* From $\beta$-smoothness, we know that $\|\nabla\widehat{L}(w)\|^2 \leq 2\beta\widehat{L}(w)$ for any $w$ (see Lemma 9 in Appendix C). Therefore, by using $\eta \leq 1/\beta$, for every $\varepsilon$,

$$\|w_{t+1} - w_\varepsilon^*\|^2 = \|w_t - \eta\nabla\widehat{L}(w_t) - w_\varepsilon^*\|^2$$

$$= \|w_t - w_\varepsilon^*\|^2 - 2\eta\langle w_t - w_\varepsilon^*, \nabla\widehat{L}(w_t)\rangle + \eta^2\|\nabla\widehat{L}(w_t)\|^2$$

$$\leq \|w_t - w_\varepsilon^*\|^2 + 2\eta\widehat{L}(w_\varepsilon^*) - 2\eta\widehat{L}(w_t) + 2\beta\eta^2\widehat{L}(w_t)$$

$$\leq \|w_t - w_\varepsilon^*\|^2 + 2\eta\widehat{L}(w_\varepsilon^*)$$

$$\leq \|w_t - w_\varepsilon^*\|^2 + 2\eta\varepsilon.$$

By summing until time $T$,

$$\|w_T - w_\varepsilon^*\|^2 \leq \|w_1 - w_\varepsilon^*\|^2 + 2T\eta\varepsilon = \|w_\varepsilon^*\|^2 + 2\eta\varepsilon T.$$

By taking a square root, using the fact that $\forall x, y \geq 0 \ \sqrt{x + y} \leq \sqrt{x} + \sqrt{y}$ and using triangle inequality,

$$\|w_T\| \leq \|w_T - w_\varepsilon^*\| + \|w_\varepsilon^*\| \leq 2\|w_\varepsilon^*\| + 2\sqrt{\eta\varepsilon T}.$$

$\square$

We rely on the following standard lemma about smooth functions (proof can be found in, e.g., 11, or in 15).

**Lemma 9.** *For a non-negative and $\beta$-smooth $f : \mathbb{R}^d \to \mathbb{R}$, it holds that $\|\nabla f(w)\|^2 \leq 2\beta f(w)$ for all $w \in \mathbb{R}^d$.*

*Proof of Lemma 2.* The proof follows the argument of [12]. First, by $\beta$-smoothness, for every $t$ and $\eta \leq 1/\beta$,

$$\widehat{L}(w_{t+1}) \leq \widehat{L}(w_t) + \nabla\widehat{L}(w_t) \cdot (w_{t+1} - w_t) + \frac{\beta}{2}\|w_{t+1} - w_t\|^2$$

$$= \widehat{L}(w_t) - \eta\|\nabla\widehat{L}(w_t)\|^2 + \frac{\eta^2\beta}{2}\|\nabla\widehat{L}(w_t)\|^2$$

$$\leq \widehat{L}(w_t) - \frac{\eta}{2}\|\nabla\widehat{L}(w_t)\|^2$$

$$\leq \widehat{L}(w_t).$$

Hence,

$$\widehat{L}(w_T) \le \frac{1}{T} \sum_{t=1}^{T} \widehat{L}(w_t). \tag{11}$$

Moreover, from standard regret bounds for gradient updates, for any $w \in \mathbb{R}^d$,

$$\frac{1}{T} \sum_{t=1}^{T} (\widehat{L}(w_t) - \widehat{L}(w)) \le \frac{\|w_1 - w\|^2}{2\eta T} + \frac{\eta}{2T} \sum_{t=1}^{T} \|\nabla \widehat{L}(w_t)\|^2.$$

By Lemma 9,

$$\frac{1}{T} \sum_{t=1}^{T} (\widehat{L}(w_t) - \widehat{L}(w)) \le \frac{\|w\|^2}{2\eta T} + \frac{\eta\beta}{T} \sum_{t=1}^{T} \widehat{L}(w_t).$$

Using $\eta \le 1/2\beta$ gives

$$\frac{1}{T} \sum_{t=1}^{T} \widehat{L}(w_t) \le \frac{\|w\|^2}{\eta T} + 2\widehat{L}(w).$$

For $w = w_\varepsilon^*$ we get by Eq. (11),

$$\widehat{L}(w_T) \le \frac{1}{T} \sum_{t=1}^{T} \widehat{L}(w_t) \le \frac{\|w_\varepsilon^*\|^2}{\eta T} + 2\widehat{L}(w_\varepsilon^*) \le \frac{\|w_\varepsilon^*\|^2}{\eta T} + 2\varepsilon.$$

$\square$

**Lemma 10.** *Let $f : \mathbb{R}^d \to \mathbb{R}$ be a $\beta$-smooth and nonnegative function. Then $f(x) \le 2f(y) + \beta\|x - y\|^2$ for all $x, y \in \mathbb{R}^d$.*

*Proof.* For any $x, y \in \mathbb{R}^d$:

$$f(x) \le f(y) + \nabla f(y) \cdot (x - y) + \frac{\beta}{2}\|x - y\|^2 \qquad (\beta\text{-smoothness})$$

$$\le f(y) + \frac{1}{2\beta}\|\nabla f(y)\|^2 + \frac{\beta}{2}\|x - y\|^2 + \frac{\beta}{2}\|x - y\|^2 \qquad (\forall c > 0 \; : \; xy \le \tfrac{1}{2c}x^2 + \tfrac{c}{2}y^2)$$

$$\le 2f(y) + \beta\|x - y\|^2. \qquad (\text{Lemma 9})$$

$\square$

# D   Proof of Section 4

## D.1   Proof of Lemma 3

**Lemma 11.** *Let $\phi$ be a tail function. Let $\ell(x)$ be the following function,*

$$\ell(x) = \begin{cases} \phi(x) & \text{if } x \ge 0; \\ \phi(0) + x\phi'(0) + \frac{\beta}{2}x^2 & \text{if } x < 0. \end{cases}$$

*Then, $\ell \in C_{\phi,\beta}$.*

*Proof.* First, it is easy to verify that $\ell$ is continuously differentiable. Second, $\ell$ is non negative: for $x \ge 0$ by the non negativity of $\phi$ and for $x < 0$ by the fact that $\phi'(0) \le 0$. Moreover, $\ell$ is convex. We need to prove that every $x < y$, $\ell'(x) \le \ell'(y)$ For $x, y < 0$, we get it by the convexity of $\phi$. For $x, y > 0$, we get it by the fact $\ell$ there is a sum of convex function and linear function. For $x < 0 < y$, by the convexity of $\phi$,

$$\ell'(x) = \phi'(0) + \beta x \le \phi'(0) \le \phi'(y).$$

In addition, $\ell$ is $\beta$-smooth. We need to prove that every $x < y$, $\ell'(y) - \ell'(x) \le \beta(y - x)$ For $x, y \ge 0$, we get it by the smoothness of $\phi$. For $x, y \le 0$, we get it by the fact that $\ell$ is a sum of $\beta$-smooth function and a linear function. For $x \le 0 \le y$, by the smoothness of $\phi$,

$$\ell'(y) - \ell'(x) = \phi'(y) - \phi'(0) - \beta x \le \beta(y - x).$$

Finally, $\ell$ is strictly monotonically decreasing. We need to prove that every $x < y$, $\ell(y) > \ell(x)$. For $x, y > 0$, we get it by the monotonicity of $\phi$. For $x < y < 0$,

$$\ell(y) = \phi(0) + \phi'(0)y + \frac{\beta}{2}y^2 \leq \phi(0) + \phi'(0)x + \frac{\beta}{2}x^2 = \ell(x).$$

For $x < 0 < y$,

$$\ell(y) = \phi(y) \leq \phi(0) \leq \phi(0) + \phi'(0)x + \frac{\beta}{2}x^2 = \ell(x).$$

$\square$

**Lemma 12.** *Let $S \sim \mathcal{D}^n$ be a sample of size n, and let $z' \sim \mathcal{D}$ be a validation example. Moreover, Assume $n \geq 35$ and let $\delta_2$ be the fraction of $z_2$ in S. We define the following event,*

$$A = \{z_3 \notin S\} \cap \{z' = z_3\} \cap \{\delta_2 \in [\tfrac{1}{32}, \tfrac{1}{8}]\}.$$

*Then,*

$$\Pr(A) \geq \frac{1}{120en}$$

*Proof.* The proof follows directly by Lemma 13 and Lemma 14. We define the following events,

$$A_1 = \{z_3 \notin S\} \cap \{z' = z_3\}, A_2 = \{\delta_2 \in [\tfrac{1}{32}, \tfrac{1}{8}]\}.$$

By Lemma 13.

$$\Pr(A_1) \geq \frac{1}{2en}.$$

By *Lemma* 14,

$$\Pr(A_2|A_1) \geq \frac{1}{60}.$$

Then, combining both results,

$$\Pr(A) \geq \Pr(A_1)\Pr(A_2 \mid A_1) \geq \frac{1}{120en}.$$

$\square$

**Lemma 13.** *Let $S \sim \mathcal{D}^n$ be a sample of size n, and let $z' \sim \mathcal{D}$ be a validation example. Then,*

$$\Pr(A_1) = \Pr(z_3 \notin S \wedge z' = z_3) \geq \frac{1}{2en}.$$

*Proof.* First, we know that,

$$\Pr(z_3 \notin S \wedge z' = z_3) = \Pr(z' = z_3) \cdot \Pr(z' \notin S \mid z' = z_3).$$

and,

$$\Pr(z' = z_3) = \frac{1}{n}.$$

Moreover,

$$\Pr(z' \notin S \mid z' = z_3) = P(z_3 \notin S) = (1 - \frac{1}{n})^n \geq \frac{1}{e}(1 - \frac{1}{n}) \geq \frac{1}{2e}.$$

Combining everything together proves our claim. $\square$

**Lemma 14.** *Assume $n \geq 35$ and let $\delta_2$ be the fraction of $z_2$ in S. Then*

$$\Pr(A_2 \mid A_1) = \Pr(\delta_2 \in [\tfrac{1}{32}, \tfrac{1}{8}] \mid A_1) \geq \frac{1}{60}.$$

*Proof.* Let $p'_i = \Pr(z_i = z_2 \mid A_1)$. Note that by the fact that $i \neq j$, $z_i, z_j$ are indepndent. Then, for every $i \neq j$, $p'_i = p'_j$. Then, by the fact that every example is independent,

$$
\begin{aligned}
p'_i &= \Pr(z_i = z_2 \mid z_3 \notin S) \\
&= \Pr(z_i = z_2 \mid z_i \neq z_3) \\
&= \frac{\Pr(z_i = z_2)}{\Pr(z_i \neq z_3)} \\
&= \frac{1}{1 - \frac{1}{n}} \Pr(z_i = z_2) \\
&= \frac{5}{64}.
\end{aligned}
$$

Then,

$$
\mathbb{E}[\delta_2 \mid A_1] = \frac{1}{n} \sum_{i=1}^{n} \Pr(z_i = z_2 \mid A_1) = \frac{1}{n} \sum_{i=1}^{n} p'_i = \frac{5}{64},
$$

and,

$$
\mathrm{Var}(\delta_2 \mid A_1) = \mathrm{Var}\left( \frac{1}{n} \sum_{i=1}^{n} 1_{\{z_i = z_2\}} \mid A_1 \right) = \frac{1}{n^2} \sum_{i=1}^{n} \mathrm{Var}(1_{\{z_i = z_2\}} \mid A_1) = \frac{5 \cdot 59}{64^2 n}.
$$

Finally, by Chebyshev's inequality, for $n \geq 35$,

$$
\begin{aligned}
\Pr(A_2 \mid A_1) &= \Pr\left( \delta_2 \in \left[ \tfrac{1}{32}, \tfrac{1}{8} \right] \mid A_1 \right) \\
&= \Pr\left( \left| \delta_2 - \tfrac{5}{64} \right| \leq \tfrac{3}{64} \mid A_1 \right) \\
&= 1 - \Pr\left( \left| \delta_2 - \tfrac{5}{64} \right| \geq \tfrac{3}{64} \mid A_1 \right) \\
&\geq 1 - \frac{64^2}{9} \mathrm{Var}(\delta_2 \mid A_1) \\
&= 1 - \frac{5 \cdot 59}{9n} \\
&\geq 1 - \frac{5 \cdot 59}{315} \\
&\geq \frac{1}{60}.
\end{aligned}
$$

$\square$

## D.2  Proof of Lemma 4

**Lemma 15.** *Let $\phi$ be a tail function. Let $\ell(x)$ be the following function,*

$$
\ell(x) = \begin{cases} \phi(x) & \text{if } x \geq 0; \\ \phi(0) + \phi'(0)x & \text{if } x < 0. \end{cases}
$$

*Then, $\ell \in C_{\phi,\beta}$.*

*Proof.* First, it is easy to verify that $\ell$ is continuously differentiable. Second, $\ell$ is non negative: for $x \geq 0$ by the non negativity of $\phi$ and for $x < 0$ by the fact that $\phi'(0) \leq 0$. Moreover, $\ell$ is convex. We need to prove that every $x < y$, $\ell'(x) \leq \ell'(y)$ For $x, y < 0$, we get it by the convexity of $\phi$. For $x, y > 0$, we get it by the linearity of $\ell$. For $x < 0 < y$, by the convexity of $\phi$,

$$
\ell'(x) = \phi'(0) \leq \phi'(y) = \ell'(y).
$$

In addition, $\ell$ is $\beta$-smooth. We need to prove that every $x < y$, $\ell'(y) - \ell'(x) \leq \beta(y - x)$ For $x, y \geq 0$, we get it by the smoothness of $\phi$. For $x, y \leq 0$, we get it by the linearity of $\ell$. For $x \leq 0 \leq y$, by the smoothness of $\phi$,

$$
\ell'(y) - \ell'(x) = \phi'(y) - \phi'(0) \leq \beta y \leq \beta(y - x).
$$

Finally, $\ell$ is strictly monotonically decreasing. We need to prove that every $x < y$, $\ell(y) > \ell(x)$. For $x, y > 0$, we get it by the monotonicity of $\phi$. For $x < y < 0$,

$$\ell(y) = \phi(0) + \phi'(0)y \le \phi(0) + \phi'(0)x = \ell(x).$$

For $x < 0 < y$,

$$\ell(y) = \phi(y) \le \phi(0) \le \phi(0) + \phi'(0)x = \ell(x).$$

$\square$

*Proof of Lemma 4.* First, we give a sketch of the proof: The proof argument is similar to that of Lemma 3. A key difference is that the example that GD classifies incorrectly does appear in the dataset (though rarely). We define a 1-Lipschitz loss function $\ell$ that decays to zero at the same rate as $\phi$, and a distribution such that there is another possible example $z_1$ and an almost "opposite" example $z_2$, that with constant probability, appears limited times in the training samples $S$. The lower bound follows from the fact that although $z_2$ appears in the dataset, the gradients of the loss function are sufficiently large so as to make the trained predictor correct on $z_2$.

For the proof, given $\gamma \le \frac{1}{8}$ and $\varepsilon \le \frac{1}{16}$, consider the following distribution;

$$\mathcal{D} = \begin{cases} z_1 := (1, 0) & \text{with prob. } 1 - p; \\ z_2 := (-\frac{1}{2}, 3\gamma) & \text{with prob. } p, \end{cases}$$

where $p = \frac{\phi^{-1}(8\varepsilon)}{72\gamma^2 T\eta}$. Note that the distribution is separable, as for $w^* = (\gamma, \frac{1}{2})$ it holds that $w^* z_1 = w^* z_2 = \gamma$. Further, consider the following loss function;

$$\ell(x) = \begin{cases} \phi(x) & \text{if } x \ge 0; \\ \phi'(0)x + \phi(0) & \text{otherwise.} \end{cases}$$

First, Lemma 15 below ensures that $\ell \in C_{\phi,\beta}$.

Next, let $S$ be a sample of $n$ i.i.d. examples from $\mathcal{D}$. Denote by $\delta_2 \in [0, 1]$ the fraction of appearances of $z_2$ in the sample $S$, and by $A_1$ the event that $\delta_2 \le 2p$. By Markov's inequality, we have $\Pr(A_1) \ge \frac{1}{2}$. Furthermore, as in the proof of Theorem 1, there exists a vector $w_\varepsilon^*$ which holds $\|w_\varepsilon^*\| \le \frac{\phi^{-1}(\varepsilon)}{\gamma}$. Then by Lemma 2 and the choice of $\varepsilon$,

$$\widehat{L}(w_T) \le 2\varepsilon + \frac{2\phi^{-1}(\varepsilon)^2}{\gamma^2} \le 4\varepsilon. \tag{12}$$

Now, we assume that $A_1$ holds. We know that

$$\delta_2 \le 2p \le \frac{\phi^{-1}(8\varepsilon)}{36\gamma^2 T\eta} \le \varepsilon \le \frac{1}{2}, \tag{13}$$

thus, conditioned on $A_1$ and by Eq. (12),

$$4\varepsilon \ge \widehat{L}(w_T) > (1 - \delta_2)\ell(w_T \cdot z_1) \ge \frac{1}{2}\ell(w_T(1)). \tag{14}$$

If $w_T(1) < 0$, we get that

$$4\varepsilon > \frac{1}{2}\ell(0) = \frac{1}{2}\phi(0) \ge \frac{1}{4}$$

which is a contradiction to our assumption that $\varepsilon \le \frac{1}{16}$. Then $w_T(1) \ge 0$ and from Eq. (14), $8\varepsilon \ge \ell(w_T(1)) = \phi(w_T(1))$. which implies that

$$w_T(1) \ge \phi^{-1}(8\varepsilon). \tag{15}$$

Now, by the fact that $\phi'(0) \le 1$ it follows that $\ell$ is 1-Lipschitz. Then, from the GD update rule,

$$w_{t+1}(2) = w_t(2) - 3\eta \cdot \gamma\delta_2\ell'(w_t \cdot z_2) \le w_t(2) + 3\gamma\delta_2\eta,$$

from which it follows that

$$w_T(2) \le 3\gamma\delta_2\eta T. \tag{16}$$

From Eqs. (13), (15) and (16) we now obtain that

$$w_T \cdot z_2 \leq 9\gamma^2 \delta_2 T\eta - \frac{1}{2}\phi^{-1}(8\varepsilon)$$

$$\leq 9\gamma^2 T\eta \frac{\phi^{-1}(8\varepsilon)}{36\gamma^2 T\eta} - \frac{1}{2}\phi^{-1}(8\varepsilon)$$

$$= -\frac{1}{4}\phi^{-1}(8\varepsilon).$$

By the fact that $\forall x < 0 : \ell(x) \geq -\frac{1}{2}x$, this implies that in the event $A_1$ it holds that:

$$\ell(w_T \cdot z_2) \geq -\frac{1}{2}w_T \cdot z_2 \geq \frac{1}{8}\phi^{-1}(8\varepsilon). \tag{17}$$

Finally, for a new validation example $z' \sim \mathcal{D}$ (independent from the sample $S$),

$$\Pr(\{z' = z_2\} \cap A_1) = \Pr(z' = z_2 \mid A_1)\Pr(A_1) \geq \frac{1}{2}P(z' = z_2) = \frac{1}{2}p \geq \frac{\phi^{-1}(8\varepsilon)}{144\gamma^2 T\eta}. \tag{18}$$

To conclude, from Eqs. (17) and (18) we have

$$\mathbb{E}[\ell(w_T z')] \geq \mathbb{E}[\ell(w_T z') \mid \{z' = z_2\} \cap A_1]\Pr(\{z' = z_2\} \cap A_1)$$

$$\geq \frac{\phi^{-1}(8\varepsilon)}{144\gamma^2 T\eta} \cdot \frac{1}{8}\phi^{-1}(8\varepsilon)$$

$$= \frac{\phi^{-1}(8\varepsilon)^2}{1152\gamma^2 T\eta}.$$

$\square$