# OpenReview forum: "Tight Risk Bounds for Gradient Descent on Separable Data"
_NeurIPS.cc/2023/Conference — NeurIPS 2023 spotlight_

### Official Review · Reviewer_GBsf · 2023-06-30

**Soundness:** 4 excellent
**Presentation:** 3 good
**Contribution:** 3 good
**Rating:** 7
**Confidence:** 3

**Summary:**

This paper considers training a convex linear model with gradient descent on linearly separable data. The paper improves the upper bound of the population risk and proves a matching lower bound.

**Strengths:**

The paper closes the gap for training convex linear models with smooth loss functions on linearly separable data, by showing matching upper and lower risk bounds. I believe this can be an important work in this field.  Also, the paper is well-written and the logic is easy to follow. The proof is a bit heavy but the authors managed to make it accessible.

**Weaknesses:**

I didn't find a major flaw in this paper. The proof looks correct to me. My only concern is that there seems to have little novelty in techniques as a theoretical work. Especially the improvement on the upper bound of the test risk relies on the results in Srebro et al. [15]. But I believe the proof of the lower bound has some nice intuition, which probably should be further explained.

**Questions:**

1. For the lower bound in Theorem 1, is there a way you can derive the minimum of the RHS with respect to the epsilon?
2. In the proof of Theorem 2, you consider two different regimes where $T \gg n$ and $n \gg T$. How to see that in Lemma 3 and 4?


**Limitations:**

As a theory paper, I think it's fine to consider linear models. Therefore, I don't see an important limitation.

---

> ### Author Rebuttal · Authors · 2023-08-09
>
> Thank you for your thoughtful comments and suggestions.  We respond in length to all of your questions below; in case there are any remaining concerns, we will be glad to clarify them during the discussion period.
>
> > Re novelty of techniques
>
> In terms of upper bounds, we believe that simplicity is an advantage of our proof approach: get stronger (and nearly tight) upper bounds using standard techniques in optimization and online learning, that improve upon more involved proof techniques used in previous works.  We think this is significant as it puts the setting of unregularized & separable linear classification back within a general and common framework, instead of analyzing it with techniques and assumptions that are tailored to the specifics of the problem.
>
> That said, we also emphasize that, to us, the main technical novelty of the paper is actually in its lower bound constructions, that are new to the literature on unregularized & separable linear classification.
>
> > “The proof of the lower bound has some nice intuition, which probably should be further explained”
>
> Agreed - we will work on including further explanations about the intuition for our final version.  Thanks for this suggestion!
>
> > “In theorem 1, is there a way you can derive the minimum of the RHS with respect to the epsilon?”
>
> Good question - for every tail function $\phi$, the $\epsilon$ which optimizes the bound is the one that satisfies the equality $(\phi^{-1}(\epsilon)^2)/\gamma^2T) = \epsilon$.  That is, the optimal $\epsilon$ for a given specific class is specified in a general yet somewhat implicit way, but at the same time, can be easily computed in a case-by-case fashion.  In Appendix A we show how to derive the bounds that Theorem 1 implies after choosing the best $\epsilon$ for a handful of loss classes.
>
> > “In the proof of Theorem 2, you consider two different regimes where T>>n, N>>T. How to see that in Lemma 3 and 4?”
>
> Lemmas 3 and 4 are based on two different hard instances of the problem. For the instance defined in Lemma 3, we get a lower bound of $\phi^{-1}(128\epsilon)^2 / \gamma^2n$, and for the instance defined in Lemma 4, we get a lower bound of $\phi^{-1}(8\epsilon)^2 / \gamma^2T$.  We only consider the relationship between $T$ and $N$ in Theorem 2, where for any regime of parameters we pick the relevant instance that establishes the lower bound (in other words, each of these lemmas is relevant for one of these regimes).

---

> > ### Comment · Reviewer_GBsf · 2023-08-17
> > **Thank you for your response**
> >
> > Thank you for addressing my questions. I will maintain my score.

---

> > > ### Author Response · Authors · 2023-08-18
> > >
> > > Thank you very much for your thoughtful response! We'll revise our paper based on the constructive feedback from the reviews.

---

### Official Review · Reviewer_GaV2 · 2023-07-02

**Soundness:** 4 excellent
**Presentation:** 4 excellent
**Contribution:** 3 good
**Rating:** 7
**Confidence:** 3

**Summary:**

This paper examines the problem of learning linearly separable data with margin using gradient descent (GD) and focuses on establishing a population risk bound on the GD output. Previous research in this area has primarily concentrated on understanding the implicit bias and population error associated with solving this task using logistic regression. In contrast, this study introduces a broader class of loss functions known as C_{\pi,\beta}, which are non-negative smooth loss functions that converge to zero more rapidly than a reference loss function, \pi.

The main contribution of this paper is the derivation of both upper and lower bounds on the population risk of GD (including stochastic gradient descent, SGD) for this class of loss functions. The authors employ a straightforward and elegant proof technique that revolves around controlling the norm of the GD output and its empirical risk. In order to establish a population error guarantee, the authors rely on uniform convergence results specifically tailored for linear models.

**Strengths:**

This paper is extremely well-written and easy to follow. Also, this paper shows that bounds on the empirical error of (S)GD and the norm of the solution suffice to provide a sharp analysis of the population error. Compared to other work in this line, proofs in this work are more straightforward. Also, the population risk guarantee in this paper can be seen as an algorithmic-dependent uniform convergence which is also interesting.


**Weaknesses:**

In classification tasks, the ultimate goal is often to minimize the 0-1 loss function, which directly measures the accuracy of classification. However, when employing gradient descent (GD) optimization, it is common to use surrogate loss functions such as logistic loss or hinge loss. These surrogate loss functions approximate the 0-1 loss and are more amenable to optimization.

In this paper, all the results focus on controlling the population risk of the surrogate loss functions. It is important to note that many surrogate loss functions can provide an upper bound on the 0-1 loss function. However, for the specific case of logistic loss, there appears to be a discrepancy between the upper bound on the logistic loss presented in this paper and the upper bound on the 0-1 loss. This discrepancy suggests that overfitting may occur, and the concept of early stopping becomes relevant.

Something that I fail to understand is why the authors do not focus on providing classification error bound for C_{pi,beta}.

**Questions:**

1- In Corollary 1, the authors instantiate their result for the logistic regression. It shows that when T is exponentially large in the number of samples, then we have overfitting. How does this result compare to Shamir 2021?


2- Can the lower bound be extended to the case that it holds for every phi and “every” loss in C_{\pi,beta}? The current statement is different.


**Limitations:**

One of the limitations is that this paper does not provide any evidence of "implicit bias".

---

> ### Author Rebuttal · Authors · 2023-08-09
>
> Thank you for your thoughtful comments and suggestions.  We respond in length to all of your questions below; in case there are any remaining concerns, we will be glad to clarify them during the discussion period.
>
> > “Why the authors do not focus on providing classification error bound for $C_{\phi,\beta}$”
>
> As you mentioned in your review, our primary focus in the paper is on the population risk rather than directly on the classification error, following the recent literature in this context (e.g., Schliserman and Koren (2022), and Telgarsky (2022)).  As we discuss in our introduction (and Schliserman and Koren (2022), and Telgarsky (2022) discuss more extensively), there are several reasons to study the loss/risk rather than the error directly; for example, it can lead to several interesting conclusions about the significance of early stopping.
>
> That said, our risk upper bound for the class of losses $C_{\phi,\beta}$ directly implies a similar bound on the (test) classification error, for all losses in this class that also upper bound the 0-1 loss (namely, whose value at zero is one).
>
> > 1. “In Corollary 1, the authors instantiate their result for the logistic regression. It shows that when $T$ is exponentially large in the number of samples, then we have overfitting. How does this result compare to Shamir 2021?”
>
> Shamir (2021) gives an upper bound for the test 0-1 error, while we prove a lower bound for the test **loss (risk)**. There is no contradiction between the two results: when running GD on the logistic loss, it is not overfitting with respect to the 0-1 error, however, it may overfit with respect to the risk. This discrepancy between the risk and the 0-1 error is actually a central aspect in our setting (see discussion in lines 83-91).
>
> > “Can the lower bound be extended to the case that it holds for every $\phi$ and “every” loss in $C_{\phi,\beta}?$“
>
> That’s a good point - the bound cannot unfortunately hold simultaneously for all losses $C_{\phi,\beta}$.  Note that $C_{\phi,\beta}$ contains all loss functions that decay to zero faster than $\phi$, and therefore contains losses that decay strictly faster than $\phi$, say faster than some $\tilde\phi \ll \phi$.  For such losses, we can obtain a better upper bound from Theorem 1 (the one for the class $C_{\tilde\phi,\beta}$), which means that the lower bound for $C_{\phi,\beta}$ cannot hold for them.
>
>
> This is often the case with lower bounds, of course: one demonstrates a hard instance of the problem within a **class of problems**, in order to conclude that stronger upper bounds for the same class of problems do not exist.
>
> > “This paper does not provide any evidence of ‘implicit bias’”
>
> In fact, our results do point to some form of “implicit bias”: the “implicit bias” in our setting holds in the form of bias to models with a small norm.  Indeed, we show that despite working in an unregularized and unconstrained regime, the effective set of possible models GD might find is relatively small (with norm bounded by a quantity depending on the tail decay rate).  Then, we can use uniform convergence arguments on this set of possible solutions to obtain generalization bounds.

---

> > ### Comment · Reviewer_GaV2 · 2023-08-16
> > **after rebuttals.**
> >
> > Thanks for addressing my questions! I raise my score to 7.

---

> > > ### Author Response · Authors · 2023-08-18
> > >
> > > Thanks a lot for your kind reply.  We will revise our paper according to the constructive comments in the reviews.

---

### Official Review · Reviewer_MaTy · 2023-07-06

**Soundness:** 3 good
**Presentation:** 3 good
**Contribution:** 2 fair
**Rating:** 5
**Confidence:** 4

**Summary:**

This paper studies the generalization properties of gradient methods for convex, smooth and decreasing losses (e.g., logistic and polynomial losses) over linearly separable data distributions.  The contributions are 1) a generalization bound based on Rademacher complexity which does not require the self-boundedness assumption for the loss and 2) a lower bound on the test loss which nearly matches the upper-bound. Overall, the results indicate that Rademacher complexity (or algorithmic stability) analyses for data-dependent generalization of GD are rate-optimal, as the lower bounds match the upper bounds in $n,T,\gamma$.

**Strengths:**

The paper is in general well-written and clear, although few inconsistencies exist throughout the main body.

 The studied problem is interesting as understanding generalization of GD on separable data is foundational to machine learning. This paper takes a step in this direction by studying linear models. The paper's results imply that the commonly studied margin-based generalization bounds which usually take the form $1/(\gamma^2 n)$ cannot be improved in $\gamma,n$, or other parameters related to tail behavior, unless additional assumptions on data are imposed. The paper's claims are new and are also applicable to a broad class of smooth losses.

The main contribution of the paper which is the lower test loss bound for a certain class of convex loss functions involves the construction of a new data distribution which can be useful for future works.

**Weaknesses:**

weaknesses and some questions:

1- The derived upper-bounds are barely an improvement over known results in literature. The authors claim in abstract and also throughout the introduction that their results apply to any smooth loss function. however this is not the case, as the loss in this work must be monotonically decreasing. This excludes the commonly used square-loss whereas previous results by Lei and Ying 2020 can be applied to square loss. I also do not find the condition of self-boundedness existing in previous works limiting as it includes almost all commonly used losses that are also used in this work. Can the authors elaborate on that?

2- Also regarding the training and generalization upper bounds, I am not convinced of the novelty of the approach. The technique based on Srebro et al. is rather simple and well-known. I assume the challenge is in characterizing the rademacher complexity for linearly separable data, but similar analyses have been already done even for more complex models such as neural networks, for example see the papers titled "Feature selection and low test error in shallow low-rotation ReLU networks" and "Polylogarithmic width suffices for gradient descent to achieve arbitrarily small test error with shallow ReLU networks".

Can the authors please comment on the challenges in optimization and generalization analyses and highlight novel steps related to general tail behaviors?

3- How do the bounds look like if $w_1\neq 0$? Particularly, it is interesting to know how the bounds on generalization behave with respect to initialization.

4- It might be helpful if a new column is added to table 2 explaining what $r_{\phi,T}$ looks like for each loss.

5- The specific data distribution used for obtaining the lower bound seems very interesting. It can be helpful for readers if the authors provide more discussion on high-level steps of the proof and the their intuition on the designed distribution.

6- It seems the bound of Theorem 1 contains a $\log ^3(n)$ factor which is missing In Table 2.

7- Table 2 shows that tailes of the form $\exp(x^{-\alpha})$ are slightly better in test loss performance than exponential tails. Does it leads to faster convergence for misclassification test error or generally for real-world experiments?

8- I think the previous works on the lower generalization bounds of SVM can be discussed here. e.g.,  "Optimality of SVM: novel proofs and tighter bounds" by S Hanneke, A Kontorovich.

**Questions:**

please see the section above.

**Limitations:**

The paper does not have a conclusions section. It can be insightful if the authors discuss some limitations of the methodology and data assumptions. There is no potential negative societal impact associated with this work.

---

> ### Author Rebuttal · Authors · 2023-08-09
>
> Thank you for your thoughtful comments and suggestions.  We respond in length to all of your questions below; in case there are any remaining concerns, we will be glad to clarify them during the discussion period.
>
> > 1. Re improvement over known results in literature
>
> First, note that monotonicity of the loss function is not a limiting condition, as any reasonable classification loss function should be monotone in its activation. This is in contrast to regression losses, like the squared loss, which are indeed not monotone (as they penalize two-sided errors).  Regarding self-boundedness, the removal of this condition helps us to remove log(T) factors for the super-exponential tail losses, compared to Schliserman and Koren (2022). Another assumption that we remove is the Lipschitz assumption, thus, we get a tight bound also for non-Lipschitz functions (e.g., the probit loss).
>
> The removal of those two conditions is also important from a theoretical perspective. The main contribution of the paper is the matching lower bound that we establish for the class of smooth functions with a given tail decay rate (what we call  $C_{\phi,\beta}$), and, there is significance in removing extraneous assumptions so that the upper and lower bounds end up matching not only in their rates but also in the set of assumptions they rely on.
>
> > 2. Re novelty of the upper bounds approach
>
> The approach we assume for our upper bound is indeed mostly simple and straightforward - we actually state and discuss this explicitly in the paper (see lines 67-77).  As we explain there, we see this as a strength since previous work in the context of classification with separable data obtained weaker results using more involved proof techniques.
>
> That said, the main step in our approach is not in characterizing the rademacher complexity — which is indeed very much standard — but rather in showing that, for a very general class of loss functions, the GD iterates remain bounded within a ball of small (tail-decay dependent) radius around initialization, and using rademacher-based uniform convergence on that ball.  Another step is using the same radius for characterizing the convergence rate of GD on the empirical risk.
> For both of these steps, it is not a priori clear why GD should converge quickly and generalize well despite being executed unconstrained on an unregularized objective, as is the case in our setting, and indeed, this was not the approach taken by much of the previous work in this domain.
>
> Please see lines 67-77 in the paper discussing this proof approach.  In light of your comments, we will try to improve this discussion in our final version.  Thanks!
>
> > 3. “How do the bounds look like if $w_1 \neq 0$?”
>
> As we explain in the paper, our upper bound is implied by two properties of the optimization algorithm - the norm of the iterates and the optimization error.  When $w_1\neq 0$, the norm of the iterate can be increased by at most $\|w_1\|$ compared the current analysis. Moreover, the optimization error can be bounded by $O(\|w_{\epsilon}^*-w_1\|^2/\gamma^2T + 2\epsilon)$.  Combining those two bounds with Proposition 1, we can obtain a test risk bound with an additional additive term that depends on $\|w_1\|^2$.
>
> We emphasize that this method is general and our bounds can be applicable to any non regularized optimization algorithm with low iterate norm and low optimization error.
>
> > 4. “new column is added to table 2 explaining what $r_{\phi,T}$ looks like”
>
> Good idea, thanks - we will revise the table as you suggest in the final version.
>
> > 5. “It can be helpful the authors provide more discussion on high level steps of the proof and the intuition on the designed distribution”
>
> Thanks for this suggestion - we will provide in the final version a more intuitive proof sketch and more high-level explanations around the construction of our lower bound instance.
>
> > 6. “$log^3(n)$ factor is missing In Table 2”
>
> Correct - in our tables we suppressed logarithmic factors other than $\log(T)$ factors  (e.g., the $log(1/\delta)$ factor is also suppressed).  We will make this clear in the revision.  Thanks.
>
>  >7.  “Are tails of the form $exp⁡(x^{-\alpha})$ better in test loss performance than exponential tails for real-world experiments?
>
> This is a very good question - we are actually not aware of empirical studies that focused on this expected behavior.
>
> > 8. “Previous works on the lower generalization bounds of SVM can be discussed”
>
> Thanks for this suggestion!  We will look into these papers and cite them accordingly in the final version.
>
> It is also worth noting that the primary focus of our work in on upper bounds and lower bounds for the (population) **risk** of GD, rather than its 0-1 classification error; thus, our lower bounds are inherently different in nature than lower bounds for the 0-1 error (e.g., the ones in "Optimality of SVM: novel proofs and tighter bounds").

---

> > ### Comment · Reviewer_MaTy · 2023-08-18
> >
> > Thank you for your response. I will maintain my score.

---

> > > ### Author Response · Authors · 2023-08-18
> > >
> > > Thank you very much for your kind reply.  We will revise our paper according to the constructive comments in the reviews.

---

### Official Review · Reviewer_5FbM · 2023-07-07

**Soundness:** 3 good
**Presentation:** 3 good
**Contribution:** 3 good
**Rating:** 6
**Confidence:** 3

**Summary:**

This paper establishes tight upper and lower bounds on the population risk of linear models trained with gradient descent (GD) on linearly separable data. In contrast to previous work, the paper's result only requires a smoothness assumption on the loss function, and the upper bounds are adaptive to the loss function's tail decay. Surprisingly, the authors achieve this result using standard techniques, with the generalization bound of [Srebro et al., 2010] based on local Rademacher complexity playing a pivotal role.

**References**

[Srebro et al., 2010]: Nathan Srebro, Karthik Sridharan, and Ambuj Tewari. Smoothness, low noise and fast rates. *NIPS* 2010.

**Strengths:**

**Minimal assumptions and standard proof technique.** The only assumptions on the loss function for the population risk upper bound are convexity, monotonicity, and smoothness, which leads to a clean theorem statement. Moreover, the proof only involves standard tools; one shows upper bounds the norm of the GD iterate and the empirical risk, and combines these two using the generalization bound from Srebro et al. (2010). In addition, the authors construct a lower bound which shows that their upper bound is nearly optimal for essentially all tail decay rates of the loss function. The minimality of assumptions, tightness of upper bounds, and simplicity of proof techniques showcases the paper’s strength.

**Weaknesses:**

**Motivation for studying losses with diverse tail decays?** Having a more general theorem is beneficial, especially if it can be achieved without much sophistication. Still, I am curious about the utility of the upper bound’s generality. Were there important examples of losses that previous works failed to capture? Are there examples of loss functions with polynomial tails that have been used in the context of classification?

**Questions:**

- What were some important losses (either from a theoretical or practical point of view) that previous works failed to cover?
- When does it make sense to use loss functions with slow (e.g., polynomial) tail decay? It seems that loss functions with slow tails may encourage the predictor to increase the margin of already correct answers at the cost neglecting misclassified samples.

**Limitations:**

Yes.

---

> ### Author Rebuttal · Authors · 2023-08-09
>
> We thank the reviewer for their thoughtful comments and suggestions. We respond in length to all of your questions below; in case there are any remaining concerns, we will be glad to clarify them during the discussion period.
>
> > Re Motivation for studying losses with diverse tail decays
>
> The relationship between properties of loss function and the generalization of gradient methods has been studied a lot recently, specifically in the setting of classification with separable data (e.g., citations [5,12] in our paper).  Our paper contributes to this line of work by demonstrating that generalization is tightly characterized by the loss function’s tail decay rate.  To support this claim,  our bounds are made very general and they capture a large variety of loss functions.
>
> > “What were some important losses (either from a theoretical or practical point of view) that previous works failed to cover?”
>
> First, the removal of the self-boundedness condition helps us to remove several log(T) factors in the super-exponential tail regime, compared to Schliserman and Koren (2022).  Second, we also remove their Lipschitz assumption and get a tight bound also for non-Lipschitz functions (e.g., the probit loss).
>
> But most importantly, the main contribution of the paper is the matching lower bound that we establish for the class of smooth functions with a given tail decay rate (what we call $C_{\phi,\beta}$). From a theoretical perspective, there is significance in removing extraneous assumptions (self-boundedness and the Lipschitz condition) so that the upper and lower bounds end up matching not only in their rates but also in the set of assumptions they rely on.
>
> > “When does it make sense to use loss functions with slow (e.g., polynomial) tail decay?”
>
> This is a great question - and one that did not yet receive a complete and satisfying answer, to the best of our knowledge.  One earlier work [5] did study polynomially tailed losses, and showed that the iterates of GD in this case converge to a direction with a non-trivial (but not maximal) margin, in contrast to losses with exponential tail, where there is convergence to max margin solution [14].  Our tight bounds show that this phenomenon is real, and in fact, polynomial tails are indeed strictly weaker than exponential tails as far as the generalization of GD is concerned.

---

> > ### Comment · Reviewer_5FbM · 2023-08-14
> >
> > Thanks for addressing my questions!

---

> > > ### Author Response · Authors · 2023-08-18
> > >
> > > Thanks a lot for your thoughtful response! We'll revise our paper based on the constructive feedback from the reviews.

---

### Decision · Program_Chairs · 2023-09-21

**Decision:**

Accept (spotlight)

**Comment:**

This paper considers generalization upper/lower bounds for linear classifiers trained with gradient methods under linearly separable distributions. Under minimal assumptions and general classes of loss functions, the paper establishes tight upper and lower bounds on the population risk.

The reviewers found the paper well-written and clear, and the reviews were in uniform support of acceptance. I also believe that the paper contains solid contributions. Some reviewers, though, raised concerns about the novelty of proofs, which should better be discussed in more detail in the revised manuscript. I agree with Reviewer MaTy that the paper will benefit from adding a conclusion section, discussing the limitations of this paper and future research directions. In the revised manuscript, please make sure to address other questions/issues raised in the reviews/discussions too.